# Functional exploration of colorectal cancer genomes using *Drosophila*

Erdem Bangi[1], Claudio Murgia[2], Alexander G.S. Teague[1], Owen J. Sansom[2] & Ross L. Cagan[1]

The multigenic nature of human tumours presents a fundamental challenge for cancer drug discovery. Here we use *Drosophila* to generate 32 multigenic models of colon cancer using patient data from The Cancer Genome Atlas. These models recapitulate key features of human cancer, often as emergent properties of multigenic combinations. Multigenic models such as *ras p53 pten apc* exhibit emergent resistance to a panel of cancer-relevant drugs. Exploring one drug in detail, we identify a mechanism of resistance for the PI3K pathway inhibitor BEZ235. We use this data to identify a combinatorial therapy that circumvents this resistance through a two-step process of emergent pathway dependence and sensitivity we term 'induced dependence'. This approach is effective in cultured human tumour cells, xenografts and mouse models of colorectal cancer. These data demonstrate how multigenic animal models that reference cancer genomes can provide an effective approach for developing novel targeted therapies.

[1] Department of Developmental and Regenerative Biology, Icahn School of Medicine at Mount Sinai, 1 Gustave L. Levy Place, Annenberg 25-40, New York, New York 10029, USA. [2] Cancer Research UK, Beatson Institute, Glasgow G61 1BD, UK. Correspondence and requests for materials should be addressed to R.L.C. (email: ross.cagan@mssm.edu).

Cancer is a genetically complex and highly heterogeneous disease with individual tumours typically carrying mutations in dozens of genes. The revolution in genome sequencing has provided unprecedented access to this genetic complexity and diversity, and paved the way for precision medicine approaches[1–3]. The basic premise of precision medicine is to identify an alteration in an 'actionable' gene, one for which an Food and Drug Administration (FDA)-approved drug or a clinical trial is available[4]. However, such actionable alterations are almost always found concurrent with other key cancer driver mutations that can differ in individual patients and tumour types. An understanding of how drug response changes in the context of complex and diverse mutation combinations will be a key to the success of precision medicine.

Genetic complexity and diversity of tumours present a major challenge for cancer drug discovery[5,6]: oncology clinical trials currently demonstrate among the lowest FDA approval rates of all major diseases[7]. With important exceptions[8–10], many drugs identified by focusing on single genes or pathways struggle to work in the clinic, even in patient populations that carry alterations in the drugs' targets. Examples include the MEK inhibitors selumetinib and binimetinib in KRAS mutant non-small cell lung cancer[11] and NRAS- or BRAF-mutated melanoma[12], respectively, as well as phosphatidyl inositol 3-kinase (PI3K) pathway inhibitors for multiple PI3K mutant tumour types[13]. Many of these clinical failures reflect the presence of concurrent mutations in the patients' genomes. Testing these agents in the context of multiple cancer driver mutations will be essential to identify genome profiles that can respond to these drugs as single agents or, potentially, to identify useful drug combinations. Recently reported methods for establishing and drug screening three-dimensional culture models derived from patients[14,15] are promising in this regard. However, exploring mechanisms of drug response and resistance, and identifying drug combinations that overcome resistance require experimental systems with sophisticated genetic tools. Mouse models built to reflect commonly observed mutation combinations in human tumours have begun to shed light on interactions between individual mutations and pathways[16,17]. Organoid tumours established by stepwise introduction of known colorectal cancer driver mutations into human intestinal cells have also recently been reported[18,19]. However, a systematic and comprehensive exploration of human cancer genomes in vertebrate model systems is a time-consuming and costly prospect, and poses a significant barrier to cancer research.

Drosophila with its arsenal of sophisticated genetic tools and long history of modelling disease is an ideal model system for functional exploration of human cancer genomes. Recent reports demonstrating a high degree of conservation of drug activity also make it a clinically relevant platform for drug screening[20,21]. To this end, we developed a diverse panel of personalized Drosophila models that reflect the multigenic and heterogeneous nature of human tumours catalogued by the Cancer Genome Atlas (TCGA)[22]. We focused on colon cancer, the second leading cause of cancer-related death in the western world. Its high mortality rate is largely due to the resistance of late-stage tumours to targeted therapies. FDA approval rates in colon cancer are 5% of drugs that enter clinical trials[7] and effective therapeutics remain an unmet need.

Highly conserved genes and pathways implicated in colon cancer have been extensively studied in cell culture and mouse models[23,24]. These pathways are highly conserved in the Drosophila hindgut, the functional equivalent of the mammalian colon[25,26]. When targeted to the adult Drosophila hindgut, our models recapitulated key features of human cancer including proliferation and disruption of the epithelial architecture, evasion of apoptosis and senescence, epithelial–mesenchymal transition (EMT), migration and dissemination to distant sites. Importantly, this work identifies intrinsic drug resistance as a key emergent property of genetically complex tumours. To demonstrate the practical utility of this approach, we identify a mechanism of resistance for the PI3K/mammalian target of rapamycin (mTOR) inhibitor BEZ235. We also identify a two-step therapy that overcomes this resistance by a novel mechanism that is also effective in cultured human colorectal cancer cell lines, xenografts and a genetic mouse model of colorectal cancer. Our work demonstrates that Drosophila can be a useful platform for rapid and large-scale functional exploration of human cancer genomes.

## Results

**Genomic analysis of colon tumours from TCGA.** Genomic studies have identified ∼25 genes recurrently mutated in colon tumours, leading to misregulation of five distinct signalling pathways[22,27–29]: Wnt, receptor tyrosine kinase (RTK)/Ras, p53, transforming growth factor-β and PI3K (Supplementary Table 1 and Supplementary Data 1). We determined the mutation status of these 25 genes within 212 TCGA colon tumours, for which there is sequencing and copy number alteration data available[22], and grouped the patients by pathway alteration status (Fig. 1a,b and Supplementary Data 2). For oncogenes, we focused only on amplification events and known activating mutations, excluding from our analysis nonsense mutations and missense mutations of unknown functional significance. For tumour suppressors, all nonsense, missense mutations and deletion events were included (Supplementary Data 1).

Consistent with previous analyses, the most frequently observed misregulation event was activation of the Wnt pathway (180 patients, 85% of the patients in the data set) followed by RTK/Ras (135 patients, 64%), p53 (131 patients, 62%), transforming growth factor-β (74 patients, 35%) and PI3K (53 patients, 25%; Fig. 1b,c). Interestingly, despite the high degree of misregulation of each pathway in the population, only a small fraction of patients had mutations exclusively in a single pathway (22 patients, 10%; Fig. 1c). Most patients have more than one well-established cancer driver mutation. Indeed, we found that 30% of the patient population carried mutations in two pathways, 40% in three pathways, 15% in four and 5% in all five (Fig. 1d). Overall, the most frequently observed event was co-misregulation of RTK/Ras, p53 and Wnt pathways.

This genetic complexity may prove significant, as most cancer drug discovery and precision medicine approaches focus on individual signalling pathways and actionable genes. If most actionable mutations in the patient population occur in the context of other cancer driver mutations, it raises a question: can tumours be best addressed as the sum of their mutations or do emergent properties alter drug response in unpredictable ways? A better understanding of drug response in a diverse set of genetically complex cancer models could inform precision medicine approaches.

**Multigenic models of colorectal tumours.** To model specific human tumours in Drosophila, we selected a transgenic Drosophila line corresponding to the most frequently mutated gene in each pathway (Fig. 1a) to represent deregulation of that pathway. We used RNA interference (RNAi) lines to knock down tumour suppressors and a $ras^{G12V}$ transgene to model the oncogenic isoform KRAS$^{G12V}$. We then generated the appropriate combination of these five transgenes to represent the pathway alteration status observed in each patient (Supplementary Table 2 and Supplementary Data 2). Our collection contains 32 multigenic models representing 212 human tumours; the exact

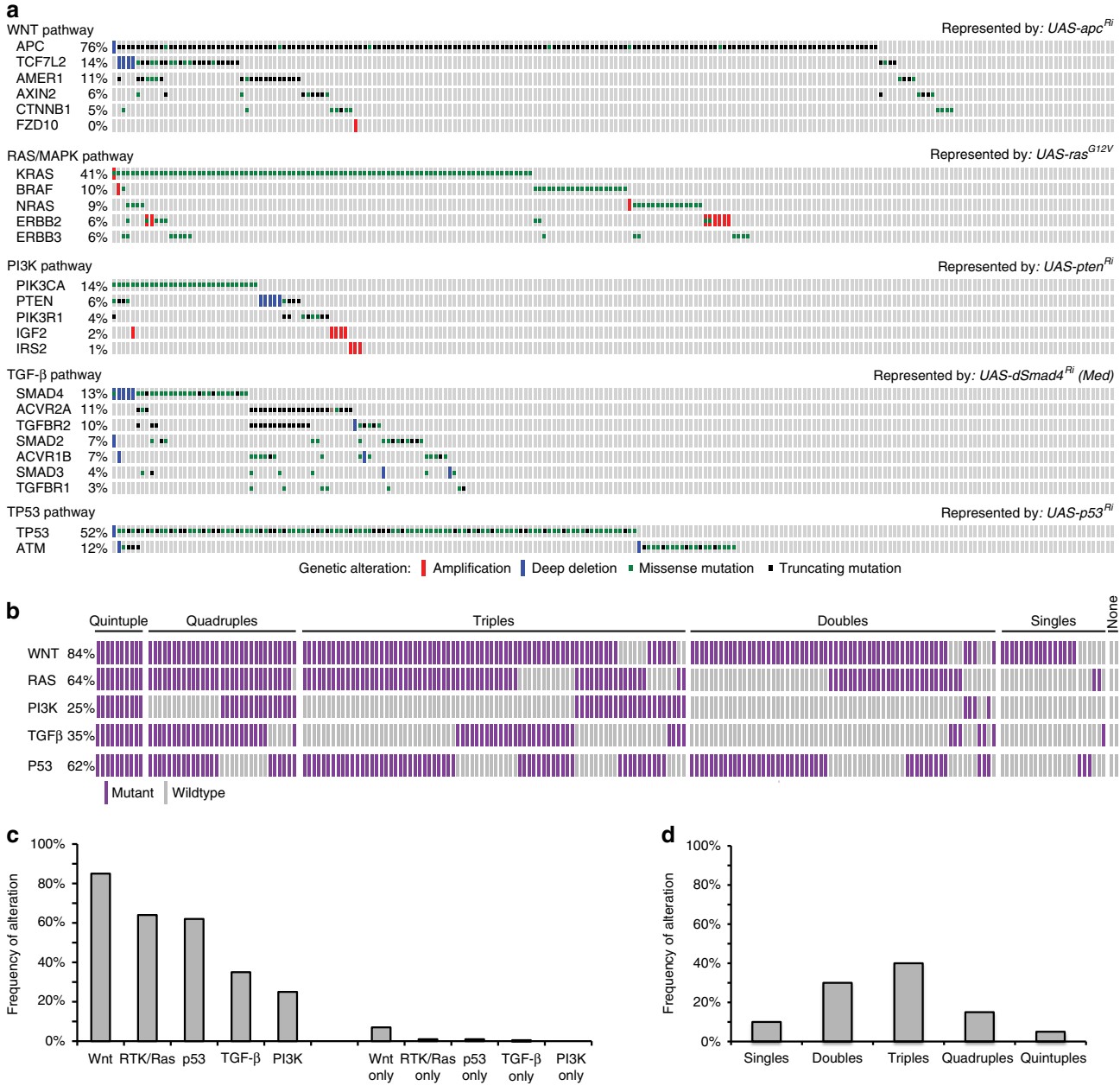

**Figure 1 | Multigenic *Drosophila* cancer models based on TCGA colorectal cancer genomes.** (**a**) Population frequencies and distribution of recurrently mutated genes within the TCGA colorectal tumour data set. Each rectangular bar indicates a patient in the population. *Drosophila* transgenic lines selected to model alteration of each pathway are indicated. (**b**) Mutation profiles of individual tumours with respect to the five deregulated pathways. Purple bars indicate patients carrying an alteration in a recurrently mutated gene within a pathway; grey bars indicate tumours with no alterations in that pathway. (**c**) Frequency of alteration of the five pathways in the patient population (left: overall frequency; right: frequency in the absence of additional alterations). (**d**) Frequency of patients that show alterations in one, two, three, four and five pathways.

mutation complement, pathway alteration status and corresponding *Drosophila* model for each human tumour is shown in Supplementary Table 2 and Supplementary Data 2.

Hallmarks of cancer include hyperproliferation, disruption of the normal tissue architecture, evasion of apoptosis and oncogene-induced senescence, migration and metastasis[30]. To determine which aspects of tumorigenesis our multigenic combinations recapitulated we targeted the appropriate transgenes specifically to the adult *Drosophila* hindgut epithelium (Fig. 2a), the functional equivalent of the mammalian colon[25,26]. The adult hindgut is a single-layer epithelium divided into three main sections along its anterior–posterior axis[25,26] (Fig. 2b): (i) the pylorus is the anterior-most region of the hindgut that

controls the passage of gut contents from the midgut to the hindgut, (ii) the ileum contains the differentiated enterocytes and (iii) the rectum sits most posteriorly.

The temperature-sensitive Gal4/Gal80$^{ts}$ system and the hindgut-specific *byn* promoter were used to control the timing and location of transgene expression[31,32]. Multigenic combinations were targeted to the adult hindgut along with Dicer2 and green fluorescent protein (GFP). Using one of the quadruple combinations *ras$^{G12V}$ p53$^{Ri}$ pten$^{Ri}$ apc$^{Ri}$*, we showed that expression levels were unaffected by the number of transgenes and that *byn-gal4* levels were sufficient to efficiently direct transgene expression and knock down (Supplementary Fig. 1).

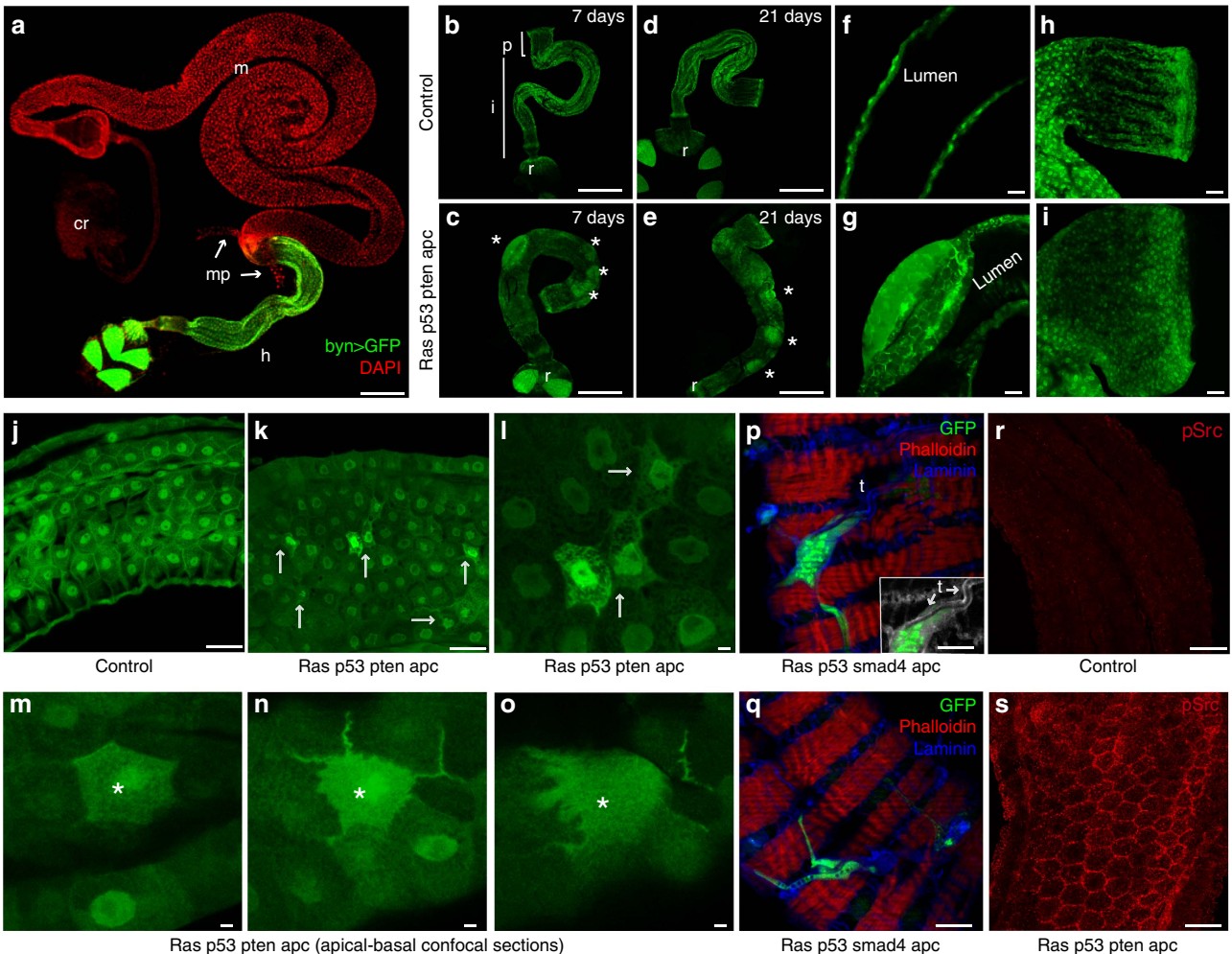

**Figure 2 | Targeting quadruple combinations to the adult hindgut.** (**a**) The adult *Drosophila* digestive track. Hindgut cells are visualized with *byn > GFP*; nuclei are in red. (**b–i**) Control (*byn > GFP,dcr2*) and *ras^{G12V} p53^{Ri} pten^{Ri} apc^{Ri}* hindguts 7 and 21 days after induction. Asterisks in **c** and **e** indicate regions of multilayering. Longitudinal optical sections (**f,g**) and pylorus regions (**h,i**) are shown. (**j–o**) Control (**j**) and *ras^{G12V} p53^{Ri}pten^{Ri} apc^{Ri}* ilea; arrows indicate migrating cells. (**l**) Close-up view view of **k**. (**m–o**) Apical-to-basal confocal sections of a migrating cell (asterisk). (**p,q**) Surface views of *ras^{G12V} p53^{Ri} smad4^{Ri} apc^{Ri}* hindguts with cells migrating on top of the muscle layer. (**p**) Inset: laminin (grey) and GFP channels only to highlight trachea (arrows). (**r,s**) Phospho-Src staining of control and *ras^{G12V} p53^{Ri} pten^{Ri} apc^{Ri}* hindguts.cr, crop; h, hindgut; i, ileum; m, midgut; mp, malphigian tubules; p, pylorus; r, rectum; t, trachea. Scale bars, 250 mm (**a–e**) and 25 mm (**f–s**).

One week after induction of the four- and five-hit models, two phenotypes became apparent: regions of multilayered epithelia formed as bulges at discrete points along the hindgut (Fig. 2b–g) and the pylorus was expanded (Fig. 2f,g), likely due to hyperproliferation. We examined our transgenic lines in various combinations to identify the primary drivers of these two phenotypes. Expansion of the pylorus was observed in response to the quintuple and all quadruples that carried *ras^{G12V}*. Notably, multilayering was observed only in animals that contained *ras^{G12V}* and *apc^{Ri}* together (see below). This indicates multilayering is an emergent property of oncogenic *ras* plus reduced *apc* activity.

**Transformed cells displayed migratory behaviour.** We also observed a strong EMT phenotype. In hindguts of four- and five-hit models—with the exception of *p53^{Ri} pten^{Ri} dSmad4^{Ri} apc^{Ri}*, which notably lacked *ras^{G12V}*—numerous cells lost their characteristic epithelial shape to assume a more mesenchymal appearance. Cells extended processes towards the basement membrane and the surrounding muscle layer (Fig. 2j–o). Further,

many epithelial cells left the hindgut epithelium to migrate on top of the surrounding muscle layer (Fig. 2p,q). Although rarely preserved during fixation, these migrating cells were commonly observed to enwrap tracheal branches, a tubular network that provides oxygen to *Drosophila* tissues (Fig. 2p, inset).

Src is a key regulator of EMT and cell migration, and its activity is commonly upregulated in advanced tumour types[33]. We assessed Src phosphorylation in *ras^{G12V} p53^{Ri} pten^{Ri} apc^{Ri}* hindguts, which displayed a strong EMT-like phenotype. Cells displayed elevated levels of phosphorylated Src that included strong membrane localization (Fig. 2r,s). Matrix metalloproteases (MMPs) are secreted by transformed cells to degrade the basement membrane during metastasis[34]; transformed fly hindguts exhibited strong but non-uniform MMP1 expression throughout the epithelium (Fig. 3a–d). Presumably, as a result, the basement membrane component Laminin was disrupted particularly in the region between the epithelium and the surrounding muscle, indicating that the integrity of the basement membrane was compromised (Fig. 3e–l). These data indicate that our colorectal cancer models recapitulate key steps in the epithelium's progression towards transformation.

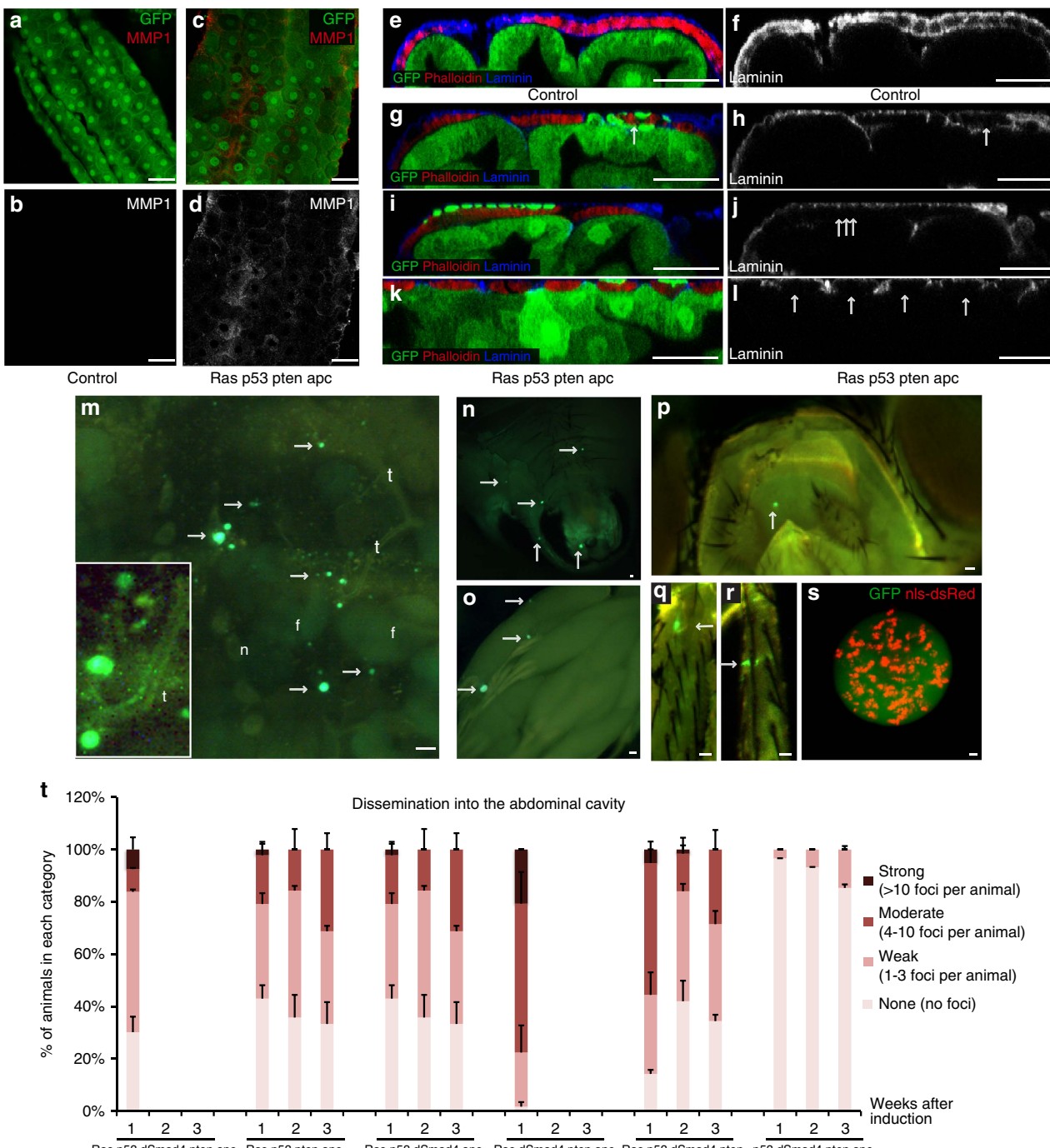

**Figure 3 | Dissemination phenotypes induced by four- and five-hit models.** (**a–d**) Mmp1 staining of control and *ras*^G12V *p53*^Ri *pten*^Ri *apc*^Ri hindguts. (**b,d**) Mmp1 channel only. (**e–l**) Cross-section views of control and *ras*^G12V *p53*^Ri *pten*^Ri *apc*^Ri hindguts. (**f,h,j,l**) Laminin channel only; arrows indicate reduced/absent laminin staining. (**m–s**) Examples of dissemination phenotype. Arrows indicate GFP-positive foci inside the abdominal cavity (**m**), underneath the abdomen epidermis (**n**), ovaries (**o**), head (**p**) and legs (**q,r**) (f, fat body, n, nephrocyte; t, trachea). (**m**) Inset: close-up view showing close association of GFP foci with tracheal branches. (**s**) Live confocal image of a multicellular GFP cluster inside the abdominal cavity. Nuclei are visualized by a nuclear dsRed transgene (nls-dsRed). (**t**) Quantification of dissemination into the abdominal cavity. Each animal is dissected and assigned into one of the following categories based on the number of disseminated foci inside the abdominal cavity: none, no dissemination; weak, 1–3 GFP-positive foci inside the abdominal cavity; moderate, 4–10 GFP-positive foci; strong, >10 GFP-positive foci (n = 2 replicates, 20–30 flies per replicate; error bars: s.e.m.). Scale bars, 25 mm.

**Transformed cells migrated to distant sites.** As GFP-positive hindgut cells transformed, many migrated to distant sites. Five- and four-hit models that contained *ras*^G12V displayed numerous GFP-positive foci throughout their bodies including the head, legs, fat body, ovaries, below the epidermis of the abdomen and within the abdominal cavity; many of these foci were multicellular (Fig. 3m–s). The disseminated foci were attached to the abdominal body wall or other organs through the tracheal system (Fig. 3m, inset), which presumably provided tracks for migrating cells to reach distant sites as well as a source of oxygen. These strong dissemination and EMT phenotypes recapitulate key early aspects of metastasis.

We quantified dissemination into the abdomen by categorizing animals based on the number of GFP-positive foci that collected into their abdominal cavity over time (Fig. 3t). A strong dissemination phenotype was evident with all combinations that contained $ras^{G12V}$ starting at 7 days after induction, with $ras^{G12V}$ $dSmad4^{Ri}$ $pten^{Ri}$ $apc^{Ri}$ displaying the strongest expressivity. We were not able to monitor dissemination beyond 1 week in this quadruple and in the quintuple model due to significant animal lethality (Fig. 3t). Time-course analysis of dissemination revealed that this phenotype did not get stronger over time in some combinations (Fig. 3t), suggesting that animals with the strongest phenotypes were progressively removed from the population. Of note, we only observed a very mild dissemination phenotype with $p53^{Ri}$ $pten^{Ri}$ $dSmad4^{Ri}$ $apc^{Ri}$ (Fig. 3t), indicating that $ras^{G12V}$ is required for significant dissemination.

Our analysis of five- and four-hit models indicated that our models capture key features of human tumours. We next explored how individual transgenes in our models interact to give rise to these tumour phenotypes. For this analysis, we focused on one of the four-hit combination lines, $ras^{G12V}$ $p53^{Ri}$ $pten^{Ri}$ $apc^{Ri}$.

**Complex gene interactions regulate transformation and migration**. The *Drosophila* hindgut epithelium is normally quiescent, although stem cells are stimulated to proliferate in response to tissue damage or $ras^{G12V}$ (Fig. 4a,b,d)[26,35]. These 5-bromodeoxyuridine (BrDU)-positive cells are the progeny of normally quiescent hindgut stem cells that are 'pushed' into the pylorus region over time[35]. We observed large numbers of BrDU-positive cells in the pylorus of $ras^{G12V}$ $p53^{Ri}$ $pten^{Ri}$ $apc^{Ri}$ hindguts labelled for 7 days, but not in controls (Fig. 4b,c). This suggests that expansion of the pylorus region in response to the quadruple combinations is also due to proliferation of normally quiescent stem cells.

$ras^{G12V}$ was necessary for this proliferation phenotype[35] (Fig. 4d): the remaining transgenes alone or in combination had no measurable effect on proliferation. Addition of $pten^{Ri}$ or $apc^{Ri}$ strongly synergized with $ras^{G12V}$ to induce proliferation and $ras^{G12V}$ $pten^{Ri}$ $apc^{Ri}$ exhibited the strongest proliferative phenotype (Supplementary Fig. 2a–d). Addition of $p53^{Ri}$ consistently reduced proliferation (Fig. 4c–e and Supplementary Fig, 2a–d); in mammals, a similar reduction has been attributed to the accumulation of persistent DNA damage over time[36,37]. Alternatively, reduced proliferation may reflect a role for p53 in sensing progressive damage within neighbouring transformed hindgut tissue.

Analysis of double and triple combinations demonstrated that multilayering required combining $ras^{G12V}$ and $apc^{Ri}$: both $ras^{G12V}$ $p53^{Ri}$ $pten^{Ri}$ $smad4^{Ri}$ and $p53^{Ri}$ $pten^{Ri}$ $dSmad4^{Ri}$ $apc^{Ri}$ lacked a multilayering phenotype (Fig. 4f–j), highlighting the importance of reduced *apc*.

We also observed complex interactions with regard to dissemination of transformed cells. $ras^{G12V}$ has been reported to induce loss of cell polarity and basal delamination in a mammalian two-dimensional monolayer cell culture model[38] and we previously observed that expressing $ras^{G12V}$ alone led to a strong hindgut dissemination phenotype[35]. We found that this $ras^{G12V}$-induced dissemination was significantly enhanced in the quadruple $ras^{G12V}$ $p53^{Ri}$$pten^{Ri}$ $apc^{Ri}$. Expressing $p53^{Ri}$, $pten^{Ri}$, $dSmad4^{Ri}$ or $apc^{Ri}$—alone or in combination—displayed low levels of dissemination (Figs 3t and 4k, and Supplementary Fig. 2k). Curiously, pairing single transgenes with $ras^{G12V}$ displayed stronger dissemination phenotypes than triple combinations or even $ras^{G12V}$ $p53^{Ri}$ $pten^{Ri}$ $apc^{Ri}$ (Fig. 4k). We therefore explored the possibility that targeting all four pathways

provides enhancements to tumour progression that outweigh reduced dissemination.

**Advantages conferred to tumours by multigenic combinations**. Migrating $ras^{G12V}$ cells—and most double combinations—were typically small and extended short processes (Fig. 4l). Migrating cells in triple and quadruple combinations were significantly larger and extended longer processes (Fig. 4l). These phenotypes suggest that triple and quadruple combinations led to a more complete transformation process, yielding apparently more robust migrating cells.

Apoptosis is an important cellular defense against cancer and tumour cells acquire methods to evade it[39]. Control hindguts displayed no detectable caspase activation as assessed by cleavage of the initiator caspase Dronc (Fig. 5a). High levels of caspase activity were observed with $ras^{G12V}$ alone, weaker levels were observed with $pten^{Ri}$ and none with $p53^{Ri}$ or $apc^{Ri}$ (Fig. 5b and Supplementary Fig. 2). In contrast to $ras^{G12V}$ alone, no caspase activation was detected in $ras^{G12V}$ $p53^{Ri}$ $pten^{Ri}$ $apc^{Ri}$ hindguts (Fig. 5c), indicating that one consequence of reducing tumour suppressor activity is to block $ras^{G12V}$-dependent caspase cleavage.

Caspase activation was reduced in $ras^{G12V}$ $pten^{Ri}$ hindguts and undetectable in $ras^{G12V}$ $p53^{Ri}$ hindguts, whereas $apc^{Ri}$ had no effect, indicating that intact Pten and P53 activities are required to initiate or sustain apoptotic signalling in the context of $ras^{G12V}$ (Fig. 5d and Supplementary Fig. 2). Further complexity emerged within triple combinations: $p53^{Ri}$ failed to suppress caspase cleavage induced by $ras^{G12V}$ $apc^{Ri}$ but did so when paired with $pten^{Ri}$ (Supplementary Fig. 2). We conclude that, in the transformation process, higher pathway complexity favours a block in apoptosis.

We observed that a subset of hindgut cells lost GFP fluorescence over time, most notably those bearing $ras^{G12V}$ or $ras^{G12V}$ $pten^{Ri}$, or, to a lesser extent, $pten^{Ri}$ alone (for example, Supplementary Fig. 2). These same GFP-negative cells were positive for the senescence marker SA-β-gal (Supplementary Fig. 2 and Fig. 3e–h), indicating emergence of at least some aspects of cellular senescence, a key cellular defense against malignant progression[40,41]. Interestingly, in hindguts carrying $ras^{G12V}$, the addition of either $p53^{Ri}$ or $apc^{Ri}$ was sufficient to reduce the number of cells positive for SA-β-gal activity (Fig. 5h and Supplementary Fig. 2). In the context of $ras^{G12V}$ $pten^{Ri}$, however, simultaneously reducing both $p53$ and $apc$ was required to prevent SA-β-gal activity (Fig. 5g and Supplementary Fig. 2). Again, greater complexity was required to promote tumour progression and evade the tissue's anti-tumour safeguards. That is, complexity favours a block in both caspase cleavage and a senescence-like phenotype, although the genetic details leading to these events are different.

Together, these results reveal emergent properties by which loci interact in complex ways to promote tumorigenesis (Fig. 5i,j). They also highlight the ability of *Drosophila* to identify these interactions within the context of the whole gut.

**Genetically complex models are resistant to targeted agents**. The success rate of colorectal cancer clinical trials has remained low as patients—in particular those with KRAS mutations—have shown resistance to a broad palate of targeted therapies. Given the genetically complex and diverse nature of colon tumours, we explored whether increasing genetic complexity influences a tumour model's response to drugs. We used dissemination into the abdominal cavity as a quantitative readout to measure compound efficacy; this provides a complex downstream phenotype that measures the ability of transformed cells to

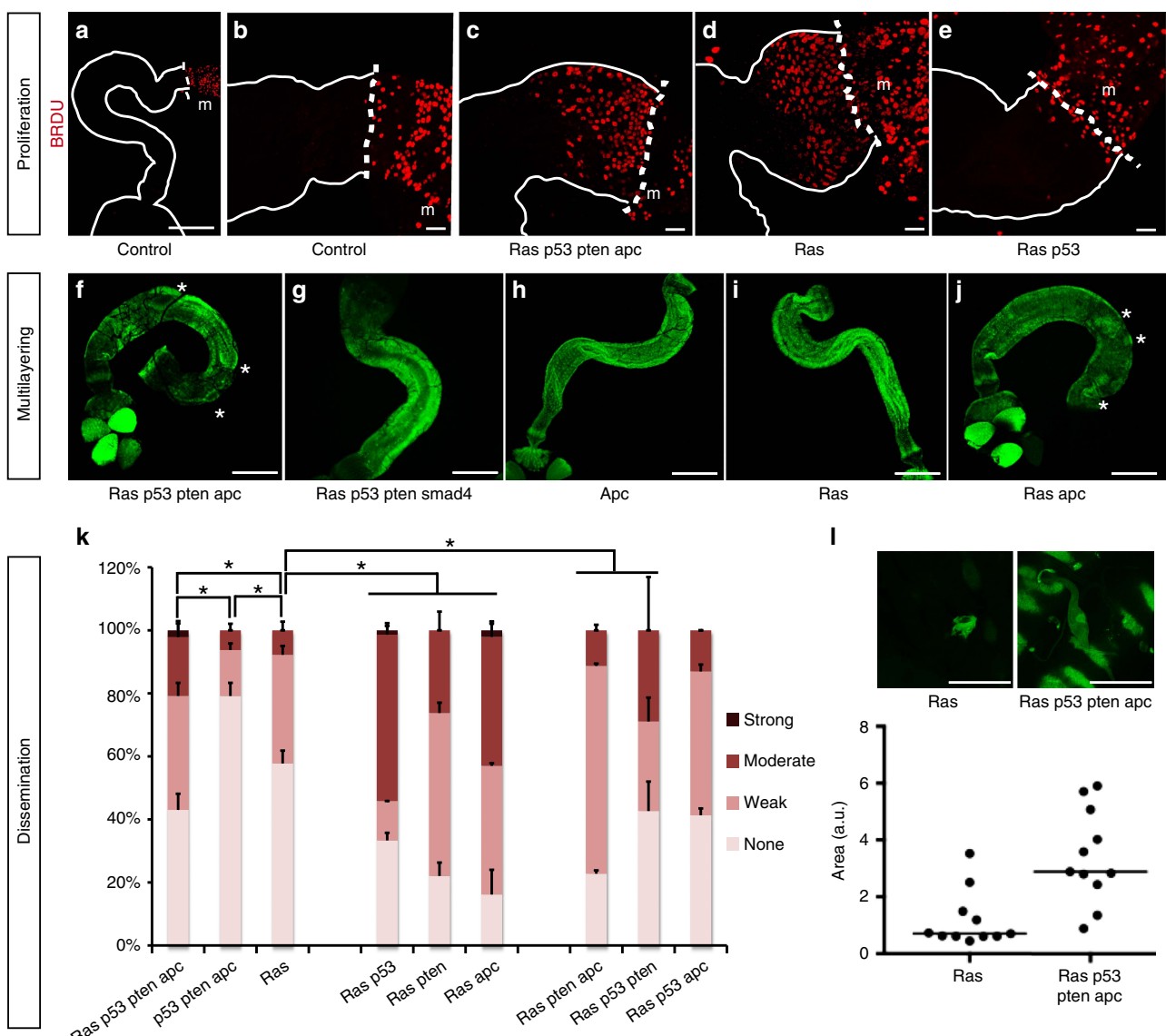

**Figure 4 | Follow-up analysis of proliferation, multilayering and dissemination phenotypes.** (**a–e**) Seven-day continuous BrDU labelling (red) of hindguts with indicated genotypes. Whole hindgut (**a**) or pylorus regions (**b–e**) are outlined with solid lines; dashed lines indicate hindgut/midgut boundary (m, midgut). (**f–j**) Whole hindguts of indicated genotypes; asterisks indicate regions of multilayering. (**k**) Quantification of dissemination one week after induction ($n = 2$ replicates, 25–30 flies per replicate; error bars: s.e.m.; *$P < 0.01$, Fisher's exact test). (**l**) Top views of $ras^{G12V}$ and $ras^{G12V}$ $p53^{Ri}$ $pten^{Ri}$ $apc^{Ri}$ hindguts with migrating cells on top and quantificaton of the migrating cell surface area. Scale bars, 250 mm (**a,f–j**), 25 mm (**b–e,l**).

migrate and survive at distant sites. We selected 16 compounds that target key cancer relevant pathways and cellular processes (Supplementary Fig. 3c); many of these compounds are currently in clinical trials, including for colorectal cancer[42,43]. Final compound concentrations in the media ranged from 0.1 μM to 1 mM; based on adult feeding rates, we estimate the amount of ingested compounds to be in the range of 10–200 ng per day per animal (Supplementary Fig. 3c). After 7 days of compound feeding, animals were dissected and number of disseminated foci in the abdominal cavity counted as described (see Fig. 3t and legend). Of note, control, $ras^{G12V}$ and $ras^{G12V}$ $p53^{Ri}$ $pten^{Ri}$ $apc^{Ri}$ all consumed similar food volumes (Supplementary Fig. 3d).

In $ras^{G12V}$ animals, 12/16 compounds significantly suppressed dissemination (Fig. 6a,b and Supplementary Fig. 3a). In contrast, the four-hit model $ras^{G12V}$ $p53^{Ri}$ $pten^{Ri}$ $apc^{Ri}$ was resistant to all tested compounds: 0/16 compounds demonstrated a statistically significant effect on dissemination (Fig. 6a,b and Supplementary Fig. 3b). We did not observe significant whole-animal toxicity,

except for the proteasome inhibitor bortezomib (velcade), which only showed efficacy at doses that otherwise killed more than 70% of the animals (Supplementary Fig. 4). The histone deacetylase inhibitor panobinostat was effective against $ras^{G12V}$ but was similarly toxic at doses that otherwise rescued $ras^{G12V}$ $p53^{Ri}$ $pten^{Ri}$ $apc^{Ri}$ (Supplementary Fig. 4). Together, our data indicate that drug resistance is an important emergent property of multigenic combinations: activating multiple cancer pathways can lead to resistance to a wide range of anti-cancer agents. We next explored a drug:tumour interaction in detail with the goal of identifying and addressing the mechanism of resistance.

**Drug resistance in multigenic combinations.** Despite the importance of PI3K pathway activity for tumour progression, inhibitors have failed to show efficacy in a broad range of tumours[44]. An example is BEZ235, the first PI3K/mTOR inhibitor to enter clinical trials[45]. Mirroring clinical trial results

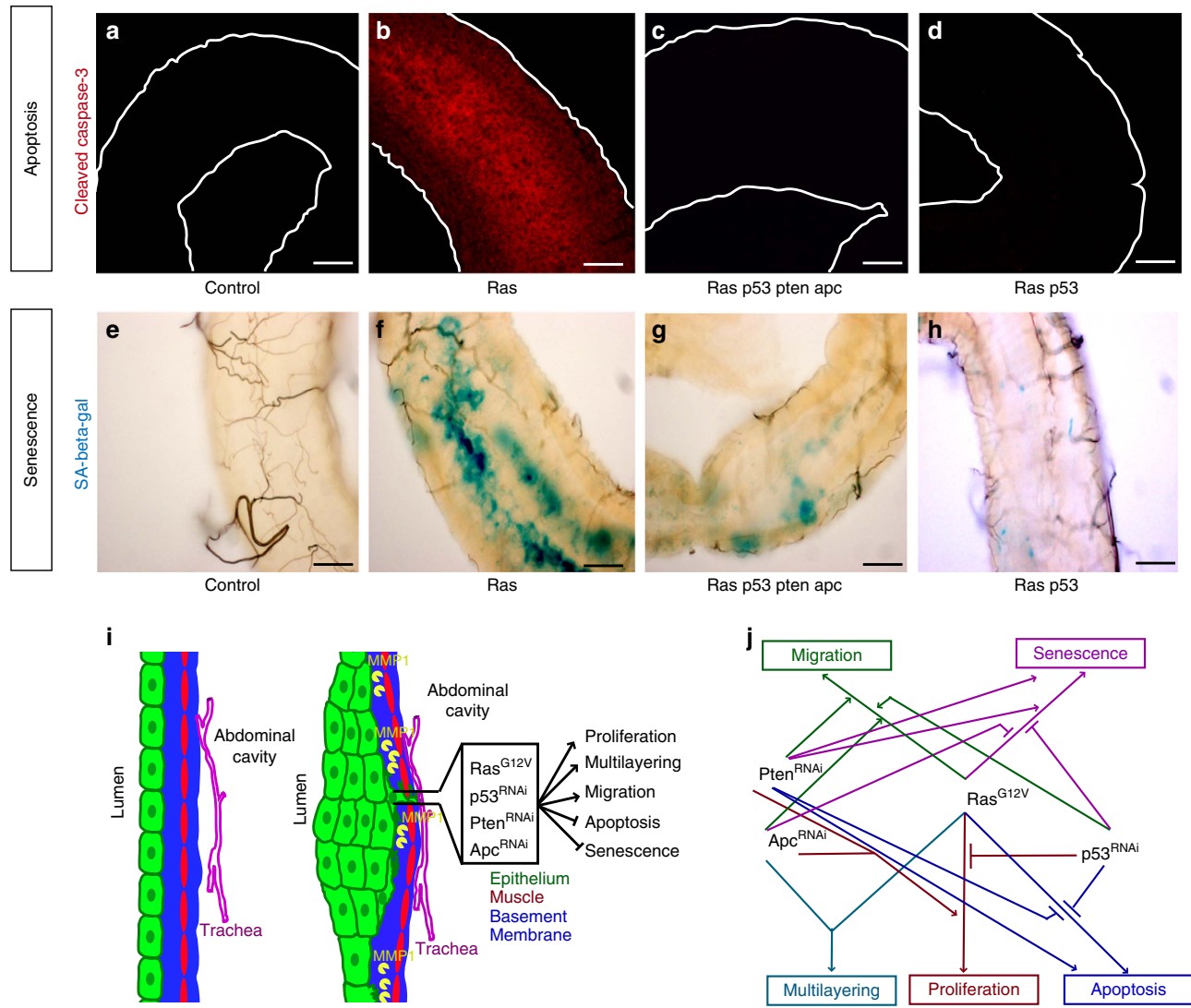

**Figure 5 | Apoptosis and senescence phenotypes in multigenic models.** Cleaved caspase-3 (**a–d**) and senescence associated (SA) β-gal (**e–h**) staining of hindguts with indicated genotypes. Hindguts outlined by solid lines in **a–d**. (**i**) Features of cancer recapitulated by our multigenic models. (**j**) Summary of interactions between individual transgenes for each phenotype. Scale bars, 25 mm.

in patients with mutations that directly activate PI3K signalling, BEZ235 failed to reduce dissemination in $ras^{G12V}\ p53^{Ri}\ pten^{Ri}\ apc^{Ri}$ animals despite their reduced phosphatase and tensin homologue (PTEN) activity. Removing $pten^{Ri}$ from the quadruple combination ($ras^{G12V}\ p53^{Ri}\ apc^{Ri}$) rendered the animals sensitive to BEZ235; conversely, pairing $ras^{G12V}$ with $pten^{Ri}$ was sufficient to direct resistance (Fig. 6c). That is, resistance to the PI3K/mTOR inhibitor BEZ235 is an emergent property of pairing $ras^{G12V}$ with reduced *pten* activity. Our results suggest that tumours with activation of both pathways may respond poorly to PI3K pathway inhibitors; recent data from preclinical models and human clinical trials also suggest BEZ235 resistance by tumours that pair activation of Ras and PI3K pathways[46–50].

PI3K activation leads to phosphorylation of AKT (p-AKT), which in turn has multiple targets such as Tor Complex 1 (mTORC1) that regulate cell survival, growth and metabolism[51,52]. Based on western analysis, $ras^{G12V}\ p53^{Ri}\ pten^{Ri}\ apc^{Ri}$ and $ras^{G12V}\ pten^{Ri}$ both had very high levels of p-AKT. However, mTORC1 activity—as assessed by 4EBP phosphorylation, a direct mTORC1 target—was low (Fig. 6d,e). This unique state of PI3K pathway output (high p-AKT/low mTORC1) was an emergent property of $ras^{G12V}$ plus $pten^{Ri}$

(Fig. 6d,e) and correlated with resistance to PI3K pathway inhibitors as single agents (Fig. 6c–e).

The high p-AKT/low mTORC1 bifurcation emerged over time. Initially, $ras^{G12V}\ p53^{Ri}\ pten^{Ri}\ apc^{Ri}$ hindguts displayed activation of both AKT and mTORC1 (Fig. 6e) but, over time, p-AKT levels steadily increased and mTORC1 activity decreased until cells reached a steady state with (i) high levels of p-AKT but (ii) low levels of p-4EBP (Fig. 6e). Previous work in mammalian cells and in *Drosophila* identified a similar high p-AKT/low mTORC1 state induced by FoxO to maintain energy homeostasis in response to physiological stress[53,54]. Furthermore, activating mutations in PI3K were associated with low TORC1 activity in estrogen receptor (ER)-positive breast tumour samples, although the status of Ras/mitogen-activated protein kinase activation was not reported[55].

This bifurcation was not observed in animals containing $ras^{G12V}$ or $pten^{Ri}$ alone. Although both AKT and mTORC1 were activated early on in response to $ras^{G12V}$ or $pten^{Ri}$ as single transgenes, by day 7 (i) p-AKT and p-4EBP levels had returned to baseline (Fig. 6d,e) and (ii) the total levels of AKT protein were very low (Supplementary Fig. 5). This altered profile likely reflects a feedback inhibitory loop that controls AKT protein stability in

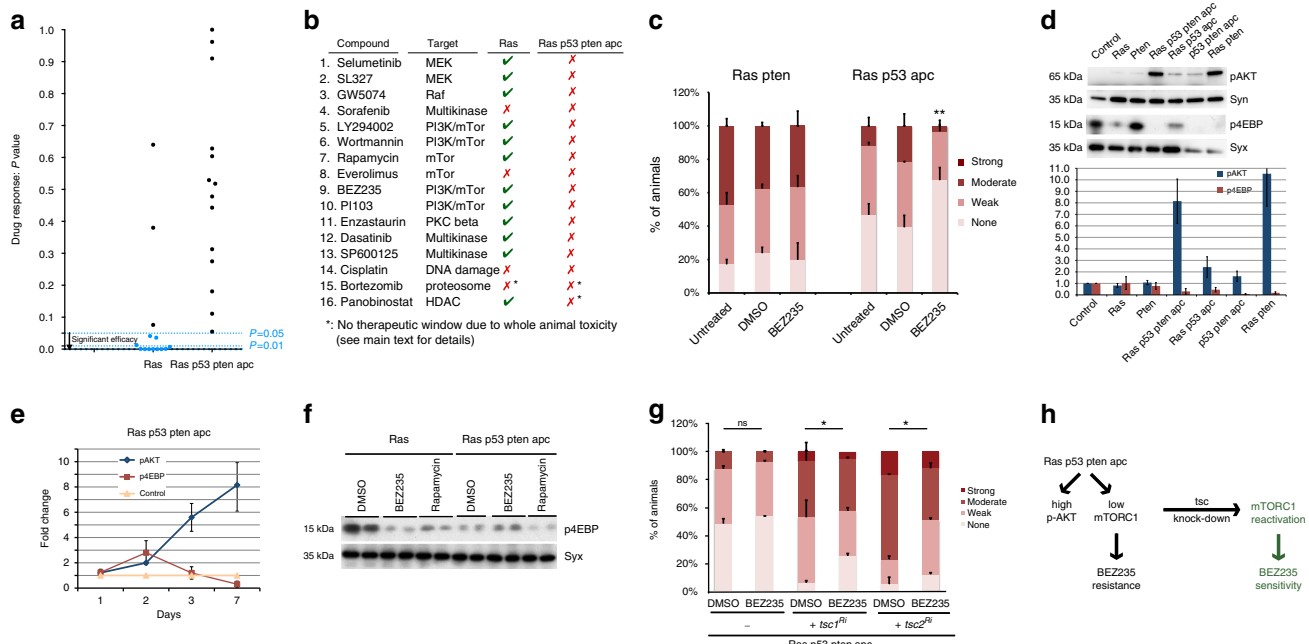

**Figure 6 | Drug resistance as an emergent feature of increased genetic complexity.** (**a**,**b**) Plot of *P*-values indicating significance of compound rescue (**a**) and summary of compound response (**b**) of *ras^{G12V}* and *ras^{G12V} p53^{Ri} pten^{Ri} apc^{Ri}* animals. *P*-values (**a**) were obtained by comparing the dissemination phenotype after compound feeding to dimethyl sulfoxide (DMSO) fed flies (dissemination plots can be found in Supplementary Fig. 3a,b). Blue dots represent statistically significant results. (**c**) Quantification of dissemination in *ras^{G12V} pten^{Ri}* and *ras^{G12V} p53^{Ri} apc^{Ri}* animals treated with BEZ235. (**d**) Western blot analysis of PI3K pathway output from hindguts with indicated genotypes 7 days after induction of transgenes and quantification. Syn, Syntaxin (loading control). (**e**) Time-course analysis of PI3K pathway activation status in control and *ras^{G12V} p53^{Ri} pten^{Ri} apc^{Ri}* hindguts. (**f**) Western blot analysis of the biochemical response by *ras^{G12V}* and *ras^{G12V} p53^{Ri} pten^{Ri} apc^{Ri}* animals to PI3K pathway inhibitors. (**g**) Quantification of dissemination in indicated genotypes treated with BEZ235 or DMSO. (**h**) Schematic illustration of the mechanism of resistance to BEZ235: genetically activating mTORC1 promotes BEZ235 sensitivity. (**d**,**e**) Each data point represents the average response of two to five biological replicates with ten hindguts per replicate; error bars: s.e.m. (**a**–**c**,**g**) *n* = 2 replicates, 30 flies per replicate; error bars: s.e.m. *P < 0.01 and **P < 0.05 (Fisher's exact test). Compound doses reflect concentrations in the food. Uncropped gels with molecular markers for **d** and **f** can be found in Supplementary Fig. 8a–c.

response to chronic PI3K pathway activation[56]. In our experiments, this feedback loop was disrupted specifically when *ras^{G12V}* and *pten^{Ri}* were present together. We conclude that sensitivity to PI3K pathway inhibitors requires high mTORC1 activity; p-4EBP levels serve as a marker for this sensitivity. Co-activation of Ras and PI3K pathways blocks mTORC1 elevation and therefore sensitivity to BEZ235 (Fig. 6h).

Consistent with this view, we observed a reduction in p-4EBP levels in response to BEZ235 and rapamycin in *ras^{G12V}* hindguts, demonstrating that these compounds targeted the PI3K pathway as expected (Fig. 6f). In contrast, p-4EBP levels were unchanged in *ras^{G12V} p53^{Ri} pten^{Ri} apc^{Ri}* hindguts. That is, PI3K pathway inhibitors reduced mTORC1 activity in *ras^{G12V}* but not in *ras^{G12V} p53^{Ri} pten^{Ri} apc^{Ri}*. This resistance likely reflects the already very low baseline levels of p-4EBP in *ras^{G12V} p53^{Ri} pten^{Ri} apc^{Ri}* hindguts (Fig. 6f). Perhaps PI3K inhibitors lose their efficacy, as they cannot reduce the mTORC1 activity lower than the already low baseline levels.

**Increasing mTORC1 activity conferred drug sensitivity.** If resistance of *ras^{G12V} p53^{Ri} pten^{Ri} apc^{Ri}* animals to PI3K pathway inhibition is due to low mTORC1 activity in trans-formed cells, activating mTORC1 in these animals should render them sensitive to PI3K pathway inhibitors. We genetically increased mTORC1 activity by reducing activity of two negative regulators of mTORC1, *tsc1* or *tsc2*. The result was a modest but statistically significant response to BEZ235 in *ras^{G12V} p53^{Ri} pten^{Ri} apc^{Ri}* animals (Fig. 6g).

Genetic manipulation experiments such as *tsc* knockdown typically generate strong long-term activation—a functional 'switch'—and lack temporal control. This is evident by the strong enhancement of dissemination by *tsc* knockdown and the modest response to BEZ235. To achieve a more transient activation of mTORC1 that does not enhance *ras^{G12V} p53^{Ri} pten^{Ri} apc^{Ri}*-induced dissemination, we used the AKT-activating compound SC79 (ref. 57). We found that SC79 also activated the *Drosophila* PI3K pathway in a dose-dependent manner (Fig. 7a). We then tested whether pharmacological activation of the PI3K pathway would confer sensitivity to subsequent treatment with PI3K pathway inhibitors in a two-step 'therapy'. Pretreatment of *ras^{G12V} p53^{Ri} pten^{Ri} apc^{Ri}* animals with SC79 followed by BEZ235 significantly suppressed dissemination, specifically at doses of SC79 that activated mTORC1 (Fig. 7b). Importantly, the efficacy of SC79/BEZ235 treatment was lost when the order of the two drugs was reversed (Fig. 7b); this rules out additive effects and emphasizes the order of drug treatment as an essential aspect of the mechanism of action of the two-step therapy. Using the most effective dose of SC79, we found that a 1-day/2-day alternating treatment schedule was also effective, whereas either drug alone did not have any statistically significant effect on dissemination (Fig. 7c). These findings support a two-step model of 'induced dependence' (Figs 6h and 8f): SC79 treatment induces dependence on mTORC1 by increasing its activity; this in turn confers sensitivity to dual PI3K/mTOR inhibitors.

These data demonstrate that a drug can be used to promote 'induced dependence' on its target pathway, creating new drug sensitivities within a tumour in a rational and controlled manner.

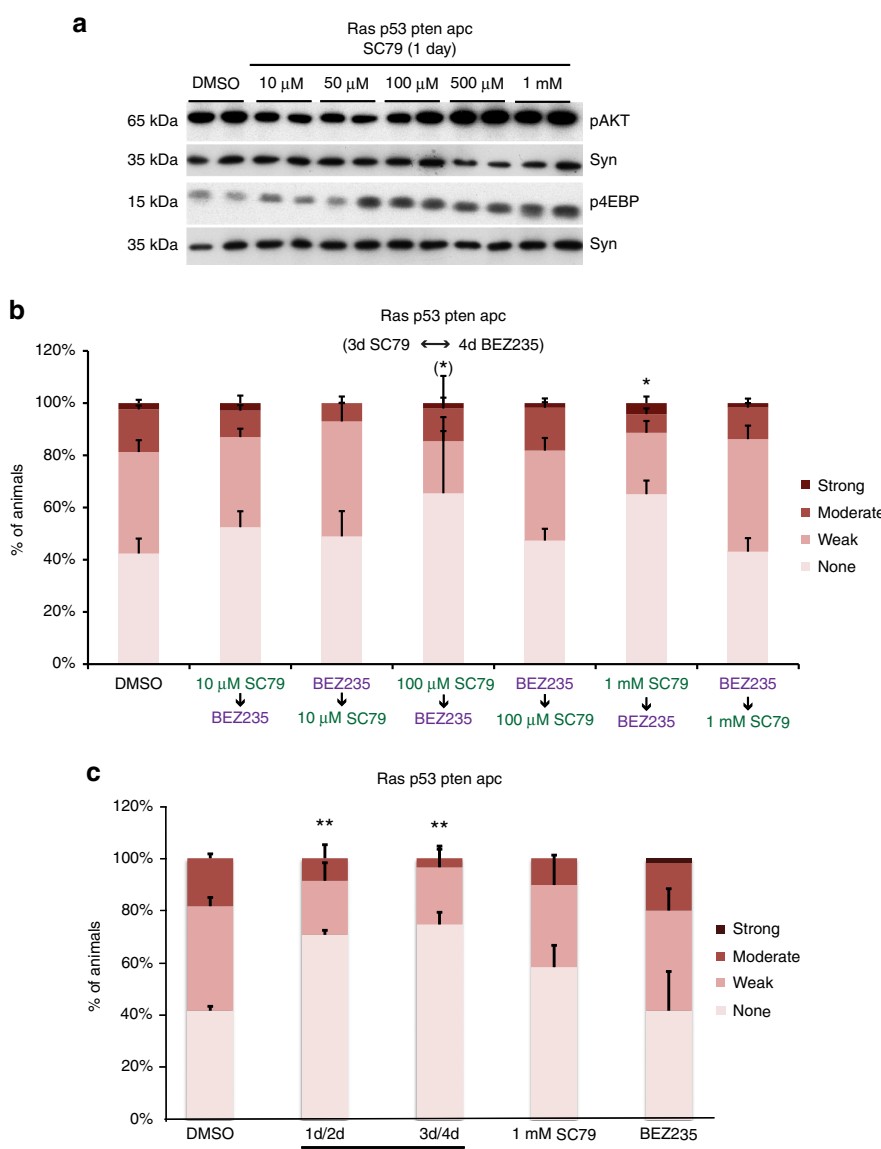

**Figure 7 | AKT activator SC79 promotes sensitivity to PI3K pathway inhibition.** (**a**) Western blot analysis of PI3K signalling pathway output in $ras^{G12V}$ and $ras^{G12V}$ $p53^{Ri}$ $pten^{Ri}$ $apc^{Ri}$ hindguts after 1 day feeding of SC79 at indicated doses. Syn, Syntaxin (loading control);ten hindguts per replicate. (**b**) Quantification of dissemination in $ras^{G12V}$ and $ras^{G12V}$ $p53^{Ri}$ $pten^{Ri}$ $apc^{Ri}$ animals after sequential treatment with BEZ235 and indicated doses of SC79. (**c**) Quantification of dissemination in $ras^{G12V}$ and $ras^{G12V}$ $p53^{Ri}$ $pten^{Ri}$ $apc^{Ri}$ animals after two different treatment schedules of SC79/BEZ235 and each drug alone. (**b,c**) $n = 2$ replicates, 30 flies per replicate; error bars: s.e.m. *$P < 0.01$ and **$P < 0.05$ (Fisher's exact test). (**b**) *Variable response; not all replicates show significant rescue. Drug doses reflect concentrations in the food. Uncropped gels used to generate panel a can be found in Supplementary Fig. 8d.

This can be useful to address the complex networks that emerge due to interactions between cancer genes. Finally, with regard to tumours with PI3K and Ras pathway mutations, mTORC1 activity is a determinant of sensitivity to PI3K/mTOR inhibitors.

**A clinically relevant two-step therapy to overcome resistance.** We had previously noted that the proteosome inhibitor bortezomib directed upregulation of mTORC1 activity. We next tested whether bortezomib can be combined with BEZ235 as a more clinically relevant two-step therapy. Bortezomib induced mTORC1 activity in $ras^{G12V}$ $p53^{Ri}$ $pten^{Ri}$ $apc^{Ri}$ animals at $1–10\,\mu M$ dose range in the food; the strongest activation was observed at $5\,\mu M$ (Fig. 8a). As with SC79, sequential treatment of $ras^{G12V}$ $p53^{Ri}$ $pten^{Ri}$ $apc^{Ri}$ animals with bortezomib followed by BEZ235 was effective in suppressing dissemination (Fig. 8b).

Reversing the order of the drugs again rendered the two-step therapy ineffective (Fig. 8b). Again, as with SC79, a 1-day/2-day alternating treatment schedule of bortezomib and BEZ235 was effective, whereas either drug alone had no effect (Fig. 8c).

Although SC79 directly and specifically activates the PI3K pathway by binding AKT, bortezomib is a proteosome inhibitor with pleiotropic effects. Our finding that reversing the order of drugs in bortezomib/BEZ235 treatment is a specific prediction of the mechanism of the action of our two-step therapy and rules out any additive effects of these two drugs. To further confirm that the ability of bortezomib to activate mTORC1 is essential for the efficacy of the two-step therapy, we genetically reduced levels of Raptor—a component of the mTORC1 complex—in $ras^{G12V}$ $p53^{Ri}$ $pten^{Ri}$ $apc^{Ri}$ hindguts. Bortezomib-induced upregulation of mTORC1 activity was blocked by knocking down *raptor* (Fig. 8d), rendering bortezomib/BEZ235 two-step therapy ineffective

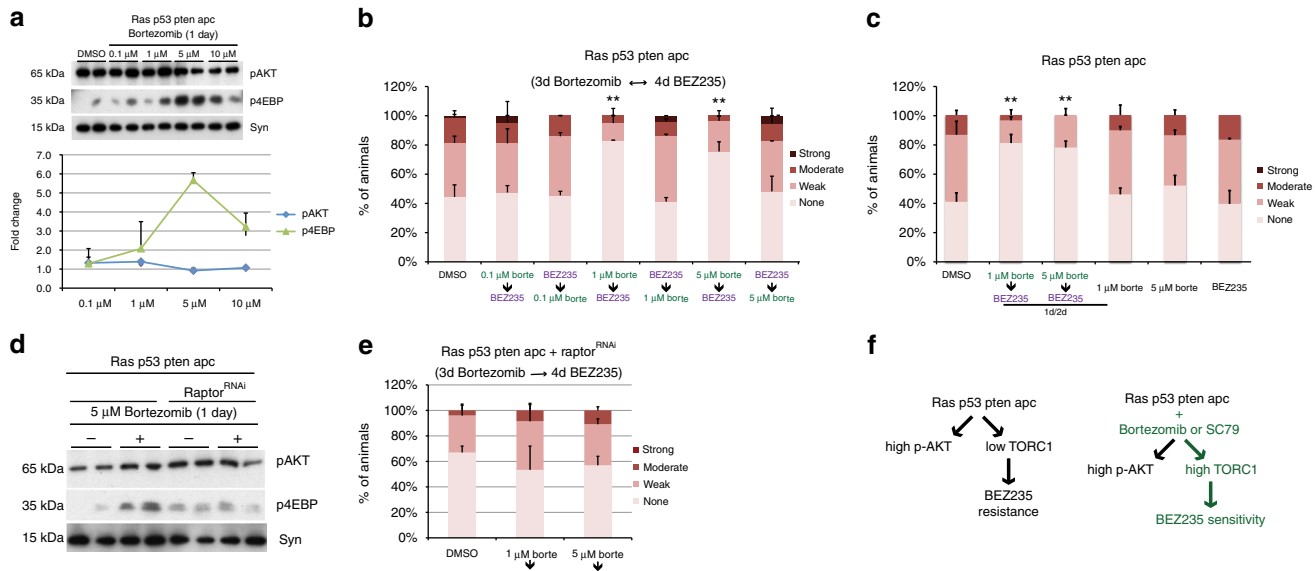

**Figure 8 | Bortezomib promotes sensitivity to PI3K pathway inhibition. (a)** Western blot analysis of PI3K pathway activity in *ras$^{G12V}$ p53$^{Ri}$ pten$^{Ri}$ apc$^{Ri}$* hindguts after 1 day feeding of bortezomib at indicated doses. Each data point represents the average response of two to five biological replicates with ten hindguts per replicate. Error bars: s.e.m. **(b)** Quantification of dissemination in *ras$^{G12V}$ p53$^{Ri}$ pten$^{Ri}$ apc$^{Ri}$* animals after sequential treatment with BEZ235 and indicated doses of bortezomib. **(c)** Quantification of dissemination in *ras$^{G12V}$ p53$^{Ri}$ pten$^{Ri}$ apc$^{Ri}$* animals after a 1-day/2-day alternating treatment schedule of bortezomib/BEZ235 and each drug alone. **(d)** Western blot analysis of PI3K pathway activity in *ras$^{G12V}$ p53$^{Ri}$ pten$^{Ri}$ apc$^{Ri}$* hindguts with and without *raptor* knockdown treated with 5 μm bortezomib for 1 day. **(e)** Quantification of dissemination in *ras$^{G12V}$ p53$^{Ri}$ pten$^{Ri}$ apc$^{Ri}$ raptor$^{Ri}$* animals after sequential treatment with indicated doses of bortezomib followed by BEZ235. **(f)** Schematic illustration of the mechanism by which the two-step therapy overcomes resistance to BEZ235: elevating mTORC1 activity increases subsequent sensitivity to BEZ235. **(b,c,e)** $n = 2$ replicates, 30 flies per replicate; error bars: s.e.m. *$P < 0.01$ and **$P < 0.05$ (Fisher's exact test). Drug doses reflect concentrations in the food. Uncropped gels used to generate panels a and d can be found in Supplementary Fig. 8e,f.

(Fig. 8e). These findings demonstrate that the efficacy of the bortezomib/BEZ235 two-step therapy is dependent on the ability of bortezomib to increase mTORC1 activity.

**Validation in mammalian models of colorectal cancer.** Finally, we examined whether our findings were relevant to human tumour cells. The human colorectal cancer cell line DLD-1 contains mutations in Ras, p53, adenomatous polyposis coli (APC) and an activating mutation in the PI3K pathway component PIK3CA, a combination that leads to an overall molecular state similar to our *Drosophila ras$^{G12V}$ p53$^{Ri}$ pten$^{Ri}$ apc$^{Ri}$* model. As our model predicted, DLD-1 proved more resistant to BEZ235 than the derivative line DLD-1-WT, in which normal PIK3CA function is restored[58] (Fig. 9a). Also consistent with our fly observations, bortezomib activated the PI3K pathway in these cells at doses and durations that did not affect survival but that rendered DLD-1 cells sensitive to BEZ235 treatment (Fig. 9b–d and Supplementary Fig. 6a,b). We confirmed these observations with the HCT116 cell line—co-activated for the Ras/mitogen-activated protein kinase and PI3K pathways—and its derivative HCT116-WT[58] (Supplementary Fig. 6c–f).

We next tested the combination therapy in DLD-1 xenografts. Bortezomib can inhibit or activate the PI3K pathway depending on the cell type, dose and duration of treatment[59–61]. Initial dosing experiments with bortezomib alone demonstrated PI3K pathway activation at a wide range of doses (0.5–0.01 mg kg$^{-1}$). Sequential treatment of xenograft tumours with bortezomib (0.01 mg kg$^{-1}$) followed by BEZ235 was significantly more effective at reducing tumour growth rates than BEZ235 alone (Fig. 9e,f), whereas bortezomib alone had no effect on tumour growth (Supplementary Fig. 7). Despite significant reduction of tumour growth rates we did not observe tumour regression, perhaps due to the more aggressive

growth rates of xenograft tumours derived from human cancer lines compared with primary human tumours[6].

Focusing on a genetically engineered mouse model, cultured spheres derived from APC/KRAS/PTEN intestinal tumours were largely insensitive to BEZ235 as a single agent; however, they responded to sequential treatment with bortezomib followed by BEZ235 (Fig. 10a). Again, reversing the order of the treatment eliminated efficacy of the two-step therapy (Fig. 10a). Furthermore, and consistent with our findings in *Drosophila*, bortezomib could be replaced by SC79 in this assay as well; reversing the order of drug administration also eliminated efficacy (Fig. 10b).

Allograft tumours derived from APC/KRAS/PTEN spheres were also sensitive to sequential treatment: strikingly, three of five allograft tumours showed regression in response to 0.01 mg kg$^{-1}$ bortezomib followed by BEZ235 (Fig. 10c). Although BEZ235 or bortezomib alone also reduced tumour size in some animals, no tumour regression was achieved (Fig. 10c). We conclude that induced dependence can be leveraged *in vivo* to reduce tumour progression.

## Discussion

As more tumours are being sequenced, there is an increasing need for model systems that provide functional validation of this sequence data at a scale and speed that can match genome-wide tumour profiling studies. We demonstrate how *Drosophila* can be used to address an important area of unmet need by modelling a major cancer type and exploring mechanisms of drug response and resistance in the context of genetically complex models. Our study is the first to build a set of genetically complex models directly based on individual tumour genome profiles. To our knowledge, these models represent the most extensive set of personalized transgenic cancer models to date.

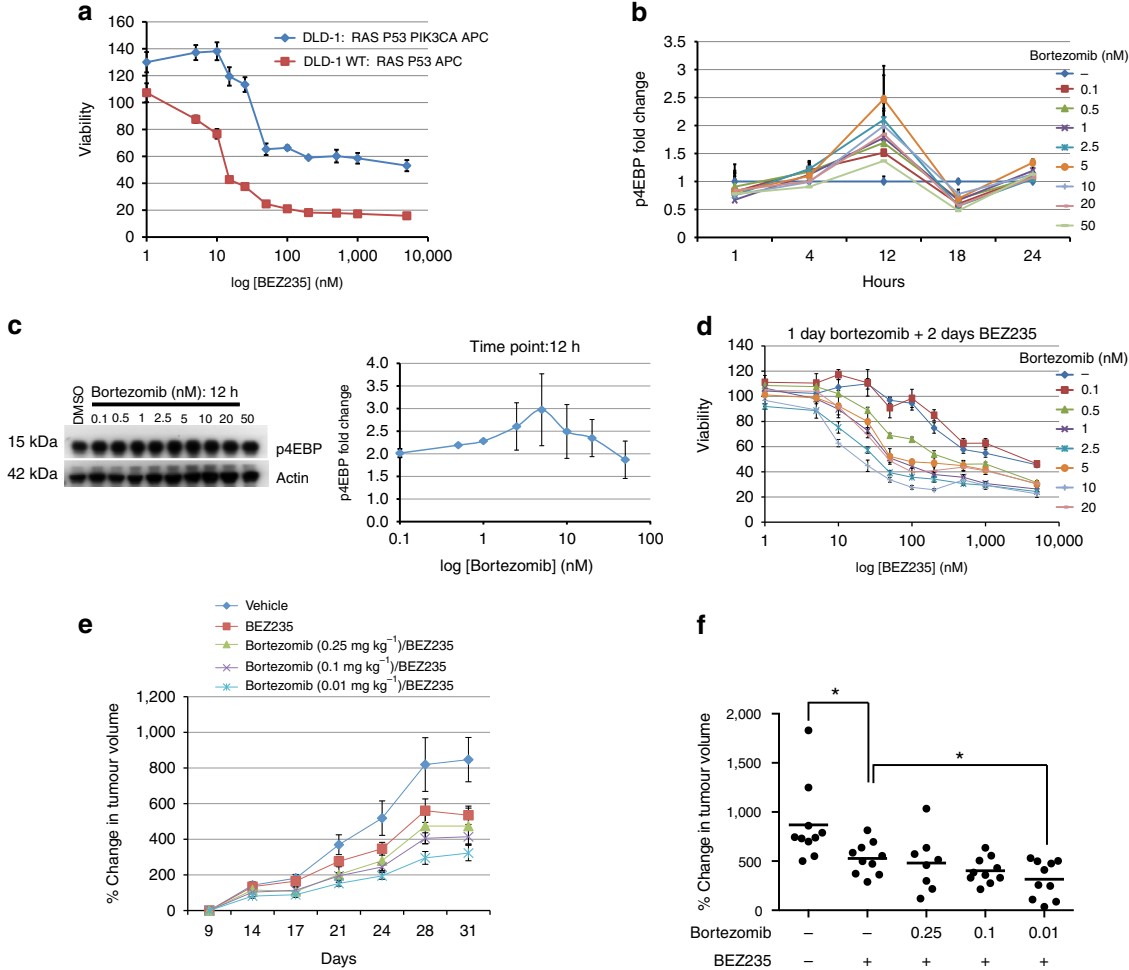

**Figure 9 | Validation of the two-step therapy in colorectal cancer cell lines.** (**a**) BEZ235 dose–response curve of DLD-1 parental (Ras and PI3K active) versus DLD-1 WT (Ras active, PI3K wild type) cell lines. (**b**) Time course of mTORC1 activation by bortezomib in DLD-1 cells. Each data point represents the average of two to three western blottings. (**c**) mTORC1 activation status in response to bortezomib 12 h after treatment. (**d**) BEZ235 dose –response curve of DLD-1 cells after pretreatment with dimethyl sulfoxide (DMSO; control) or indicated doses of bortezomib for 24 h. (**e**) Growth rates of DLD-1 xenografts treated with indicated compounds. Data points are normalized to tumour size at the beginning of drug treatment (day 9). (**f**) Percent change in DLD-1 xenograft tumour volumes at the last day of treatment (day 31; $P < 0.05$, Mann–Whitney test). Error bars: s.e.m. 10 animals. Uncropped gels used to generate **c** can be found in Supplementary Fig. 8g.

Our multigenic models capture key features of human colon tumours many of which showed complex multigenic regulation (Fig. 5i,j). The significant variation observed between specific transgene combinations emphasizes the potential differences between tumours from different patients and suggests the importance of personalized therapies. For example, we show that resistance to targeted therapies is a key emergent feature of genetically complex tumours. If drug response is highly dependent on genomic context with different mechanisms of resistance depending on the identity of concurrent driver mutations, more sophisticated patient stratification approaches based on whole-genome profiles rather than individual genes will be key to the success of precision medicine approaches. Such a transition from actionable genes to actionable genomes can only be accomplished by using more diverse and personalized preclinical models for cancer drug discovery. Such models can also lead to the discovery of a different class of anti-cancer agents that target novel emergent vulnerabilities that are revealed as a result of complex interactions between concurrent mutations.

Exploring the mechanisms of drug response and resistance, we found that resistance to PI3K pathway inhibitors such as BEZ235 was an emergent property of Ras activation plus Pten

loss. These results match recent data from multiple preclinical models and from human clinical trials as well[46,47], demonstrating that *Drosophila* can be a useful platform to identify mutation profiles associated with drug resistance in cancer patients and identify underlying molecular mechanisms. PI3K pathway is one of the most investigated pathways for cancer therapy, with a large number of therapeutic agents under clinical development. Despite the importance of this pathway in oncogenic transformation, PI3K pathway inhibitors have demonstrated modest clinical activity as single agents[44]. Our demonstration that the resistance state of $ras^{G12V}$ $pten^{Ri}$ cells can be altered by genetically increasing baseline levels of mTORC1 activity opens up new avenues for PI3K pathway inhibitors in the clinic as part of drug combinations that exploit this mechanism of resistance. Furthermore, the two-step therapy was also effective in cell lines that carry activating mutations in PIK3CA, rather than PTEN loss. Despite the well-established differences in PTEN loss and PIK3CA activation, these mutations behave similarly in their response to the two-step therapy, suggesting that tumours carrying either PTEN loss or oncogenic mutations in PIK3CA could benefit from the two-step therapy. Our demonstration that bortezomib can alter the resistance state of $ras^{G12V}$ $pten^{Ri}$

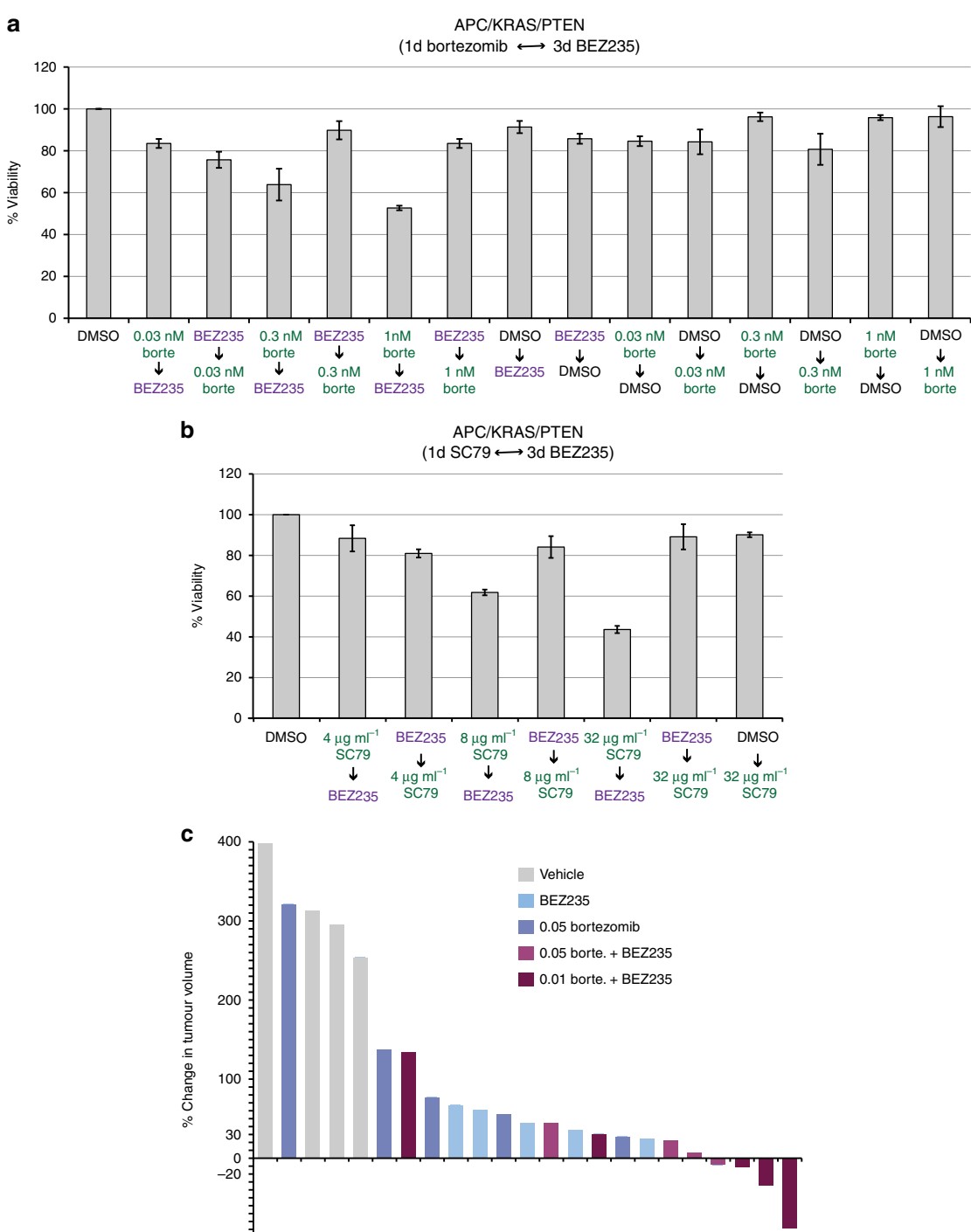

**Figure 10 | Validation of the two-step therapy in three-dimensional colosphere cultures and allografts.** (**a,b**) Cell viability of sphere cultures derived from colon tumours of an *APC/KRAS/PTEN* mouse model of colorectal cancer treated with bortezomib (**a**) or SC79 (**b**) followed by BEZ235. (**c**) Waterfall plot demonstrating the percent change in tumour volumes of allografts derived from cultured *APC/KRAS/PTEN* colospheres at the end of treatment. (**a,b**) Error bars represent s.e.m.; $n = 3$ replicates.

cells—induced dependence—suggests that combinatorial therapy or the use of less selective compounds is a useful approach against at least some genetically complex tumours.

## Methods

**Tumour genome analysis.** Mutation profiles of 212 TCGA colorectal tumours—for which whole-exome sequencing and copy number alteration data are available—were analysed using cBioPortal[62,63]. Genomic analysis was focused on 25 genes previously identified by TCGA as recurrently mutated genes in colorectal cancer.

**Fly strains and multigenic models.** Fly lines used in this study are (chromosome locations and sources in parentheses): UAS-*ras*[G12V] (second, G. Halder), UAS-*p53*[Ri] (second, VDRC), UAS-*pten*[Ri] (third, VDRC), UAS-*apc*[Ri] (second and third, VDRC), UAS-*dSmad4*[Ri] (second and third, VDRC), UAS-*tsc1*[Ri] (second, TRIP), UAS-*tsc2*[Ri] (second, VDRC) and UAS-*raptor*[Ri] (second, TRIP), UAS-*dcr2* (X, Bloomington), UAS-*GFP* (third, Bloomington), *byn-gal4* (third, V. Hartenstein) and *tub-gal80*[ts] (third, Bloomington). Double, triple and quadruple models were generated by making the following double recombinant chromosomes and combining them with each other or individual transgenes using standard genetic crosses; of note, the complexity of the multi-transgene lines prevented backcrossing and we cannot exclude effects from differing genetic backgrounds. On the second

chromosome: $UAS$-$ras^{G12V}$ $UAS$-$p53^{Ri}$, $UAS$-$ras^{G12V}$ $UAS$-$apc^{Ri}$ and $UAS$-$p53^{Ri}$ $UAS$-$apc^{Ri}$. On the third chromosome: $UAS$-$dSmad4^{Ri}$ $UAS$-$apc^{Ri}$, $UAS$-$dSmad4^{Ri}$ $UAS$-$pten^{Ri}$ and $UAS$-$pten^{Ri}$ $UAS$-$apc^{Ri}$. For the quintuple, a triple recombinant second chromosome containing $UAS$-$ras^{G12V}$, $UAS$-$p53^{Ri}$ and $UAS$-$apc^{Ri}$ was generated and combined with the third chromosome containing $UAS$-$dSmad4^{Ri}$ and $UAS$-$pten^{Ri}$ transgenes. All models contain the same second chromosome insertions of $UAS$-$ras^{G12V}$ and $UAS$-$p53^{Ri}$, and third chromosome insertions of $UAS$-$apc^{Ri}$, $UAS$-$dSmad4^{Ri}$ and $UAS$-$pten^{Ri}$ with the following exceptions: the quintuple $ras^{G12V}$ $p53^{Ri}$ $dSmad4^{Ri}$ $pten^{Ri}$ $apc^{Ri}$, the quadruple $ras^{G12V}$ $apc^{Ri}$ $dSmad4^{Ri}$ $pten^{Ri}$ and the triple $apc^{Ri}$ $dSmad4^{Ri}$ $pten^{Ri}$ carry a second chromosome insertion of $UAS$-$apc^{Ri}$.

**Targeted expression in the adult hindgut.** Multigenic combinations were targeted to the adult hindgut using the hindgut-specific GAL4 driver line $byn$-$gal4$ (V. Hartenstein) plus $tub$-$gal80^{ts}$ (Bloomington). Adult females used in this study were generated by crossing flies carrying these multigenic combinations to $UAS$-$dcr2$; $+$; $byn$-$gal4$ $UAS$-$GFP$ $tub$-$gal80^{ts}/S$-$T$. Crosses were kept at 16 °C, to keep the transgenes silent during development, and adult females were transferred to 29 °C, to induce transgene expression. In addition, GFP and Dcr2 (Dicer2) were co-expressed to mark the hindgut cells and to facilitate RNAi-mediated knockdown, respectively.

**Compound feeding.** Compounds used in this study were (stock concentrations and sources in parentheses) selumetinib (200 mM, Selleck Chemicals), SL327 (200 mM, Tocris), GW5074 (100 mM, Tocris), sorafenib (200 mM, LC Labs), LY294002 (200 mM, LC Labs), rapamycin (200 mM, LC Labs), BEZ235 (20 mM, LC Labs), dasatinib (200 mM, LC Labs), SP600125 (200 mM, LC Labs), bortezomib (200 mM, LC Labs), cisplatin (200 mM, Sigma), panobinostat (200 mM, LC Labs), wortmannin (200 mM, LC Labs), PI-103-HCl (100 mM, Tocris), everolimus (100 mM, LC Labs), enzastaurin (50 mM, LC Labs) and SC79 (200 mM, Tocris).

Compound-treated food was made by diluting compound stocks in semi-defined $Drosophila$ medium to 0.5% dimethyl sulfoxide. Adult flies were provided with fresh compound/food every other day. Animals were randomized before drug treatment and scored blindly. Animals that were dead at the time of scoring were excluded. Food intake was quantified 4 days after induction for a period of 8 h using the CAFÉ assay as previously described[64,65].

**Statistical analysis of the dissemination assay.** Sample sizes and phenotypic categories for the dissemination phenotype were determined empirically to generate a robust and reproducible quantitative readout. Animals that were dead at the time of scoring were excluded. Statistical significance for the dissemination assay was calculated using Fisher's exact test, to allow the analysis of contingency tables where sample sizes are relatively small. Use of a $4 \times 2$ contingency table allowed pairwise comparisons of the dissemination phenotype, which has four phenotypic categories, between different genotypes and/or treatment conditions. Each experiment was performed in duplicate ($n = 25$–$30$ for each replicate) and multiple times. Error bars indicate s.e.m. Experiments that show a high degree of variance between biological replicates were considered inconclusive and repeated.

**Immunohistochemistry.** Hindguts dissected in ice-cold PBS were fixed with 4% paraformaldehyde, 0.3% Triton X in PBS for 15 min, rinsed 3 × in PBS, washed 15 min in PBS, blocked 1 h in 0.1% Triton X, 1% normal goat serum in PBS, incubated with primary antibody at 4 °C overnight, rinsed three times in PBS, blocked for 1 h and incubated in secondary antibody for 2 h at room temperature. Hindguts were mounted in Vectashield with 4,6-diamidino-2-phenylindole (Vector Laboratories). Primary and secondary antibodies were diluted in block solution.

Primary antibodies were as follows: rabbit anti-Laminin (1:500, Abcam, catalogue number ab47651), mouse anti-MMP1 (1:100, DSHB, catalogue number 3B8D12-s), rabbit anti-Src-pY418 (1:100, Invitrogen, catalogue number 44660G), mouse anti-BrDU (1:10, BD Biosciences, catalogue number 347580), rabbit anti-APC (1:1,000, gift from Y. Ahmed[51]) and rabbit anti-cleaved Caspase 3 (1:100, Cell Signaling, catalogue number 9661S). Alexa-568- or 633-conjugated goat anti-mouse and anti-rabbit antibodies were used as secondary antibodies (1:1,000, Invitrogen catalogue number A-11036, A-11004, A-21126 and A-21071). Muscle labelling was performed with Alexa-568-conjugated phalloidin (1:200, Invitrogen, catalogue number A-12380). SA-β-gal staining was performed using a kit from Cell Signaling (catalogue number 9860).

**BrDU incorporation assay.** To label proliferating cells, adults were fed fed 5 mg ml$^{-1}$ BrDU in semi-defined $Drosophila$ medium daily for 7 days. Dissected guts were processed for antibody staining as described above with the addition of an initial DNAse treatment step (200 U ml$^{-1}$) at 37 °C for 1 h.

**Imaging and scoring.** Fluorescence images were visualized on a Leica TCS SPE confocal microscope and processed using Leica LAS-AF software. Scoring and imaging of SA-β-gal staining was performed using an Olympus BX41 microscope and a Nikon DS-Ri1 camera. Dissemination was imaged and scored using a Leica MZ16F dissecting scope with a GFP filter under × 10 magnification.

**Western blot analysis.** Lysates for western blot analysis were made by grinding ten hindguts in lysis buffer (50 mM Tris-HCl, 150 mM NaCl, 1 mM EDTA and 1% NP-40) supplemented with protease and phosphatase inhibitor cocktails (Sigma). Lysates were boiled for 10 min in SDS sample buffer and reducing agent, resolved on SDS–PAGE and transferred using standard protocols. Protein concentrations were determined using BioRad Protein Assay Dye Reagent. For signal detection, Immobilon Chemiluminescent horseradish peroxidase Substrate (Millipore) was used. After blocking, membranes were cut into horizontal strips and probed with appropriate antibodies; membranes were reassembled before developing. Bands were quantified using Gel Analyzer. Uncropped gels used to assemble the panels shown in all figures along with molecular markers are shown in Supplementary Fig. 8.

Primary antibodies were as follows: rabbit anti Drosophila phospho-AKT (p-Ser505; $Drosophila$ equivalent of mammalian AKT p-Ser473, 1:1,000, Cell Signaling, catalogue number 4054S), rabbit anti-mouse AKT (1:1,000, Cell Signaling, catalogue number 4691S), rabbit anti-mouse phospho-4EBP (Thr37/46, 1:1,000, Cell Signaling, catalogue number 2855S), rabbit anti-human phospho-AKT (Ser473, 1:1,000, Cell Signaling, catalogue number 4060S), mouse anti-chicken α-actin (1:1,000, DSHB catalogue number JLA20-s), rabbit anti-GFP (1:2,000, Sigma, catalogue number G1544), rabbit anti Drosophila P53 (1:1,000, DSHB, catalogue number H3), rabbit anti Drosophila Pten (1:1,000, gift from A. Wodarz[52]), mouse anti-dpERK (Thr183/Tyr185, 1:2,000, Sigma, catalogue number M8159), rabbit anti human phospho-S6 (Ser 235/236, 1:1,000, Cell Signaling, catalogue number 2211) and mouse anti Drosophila Syntaxin (1:1,000, DSHB, catalogue number 8C3). Horseradish peroxidase-conjugated anti-mouse and anti-rabbit secondary antibodies were used (1:5,000, Cell Signaling, catalogue number 7074, 7076).

**Cell culture.** Parental DLD-1 and HCT-116 cell lines (ATCC) and their PI3K wild-type derivatives (DLD-1 WT and HCT116 WT)[58] from Dr. Bert Vogelstein's laboratory were maintained using DMEM medium with 10% fetal bovine serum (FBS). As previously reported[58], PI3K pathway activity was sensitive to serum level and batch; all experiments were done under low serum conditions (2.5% FBS, CellGro, Lot number K043-6HI).

**MTT assay.** Cells were grown in sterile 25 cm$^2$ flasks to 80% confluence, trypsinized, re-suspended in 100 ml DMEM with 2.5% FBS and seeded in 96-well plates in equal numbers. After 24 h, cells were treated with compounds diluted in DMEM with 2.5% FBS as indicated. MTT (3-(4,5-dimethylthiazol-2-yl)-2, 5-diphenyltetrazolium bromide) assay was performed by replacing medium with fresh DMEM plus 2.5% FBS and 10 mg ml$^{-1}$ MTT reagent (Fisher Scientific) in PBS at a ratio of 5:1 (120 μl per well; 96-well plates). Cells were incubated at 37 °C for 3 h, media removed and cells incubated in MTT solvent (4 mM HCl, 0.1% NP40 in isopropanol) on a shaker. MTT formazan levels were determined spectro-photometrically. Each experiment was done in quadruplicate and relative viability expressed as fold change in adjusted absorbance compared with untreated and dimethyl sulfoxide-treated controls in each plate. Cell lines were checked for mycoplasma contamination before use.

**Xenograft study.** Xenograft experiments were carried out by Gregory Carbonetti and Dr. Elisa de Stanchina at the Antitumor Assessment Core Facility, Memorial Sloan-Kettering Cancer Center in compliance with institutional guidelines under an institutional animal care and use committee (IACUC)-approved protocol. Nu/nu athymic females were injected with ten million cells subcutaneously (single flank) with matrigel. Drug treatment was initiated when tumours reached ∼80–100 mm$^3$ using the following schedule: Monday: bortezomib (intraperitoneal) or vehicle; Tuesday–Friday: BEZ235 (per os, 40 mg kg$^{-1}$ per day); Saturday and Sunday: off for 3 weeks. Tumour growth was measured twice per week. Statistical significance was determined using Mann–Whitney test ($P < 0.05$). 6–8 weeks old mice were used in experiments.

**Colon sphere cultures and allograft experiments.** Adenomas from $APC/KRAS/PTEN$ mouse tumours were cultured in vitro as described[66]. Spheres were cultured for 3 days and treated with indicated doses of bortezomib, SC79 and BEZ235. Cell viability was quantified using CellTiter-Blue cell viability assay (Promega) following the manufacturer's protocol. Drug treatment was done without epidermal growth factor. For allograft experiments, five spheres were injected subcutaneously into each mouse (CD1 nude mice), tumours were established for 10 days before treatment: Monday: bortezomib (intraperitneal) or vehicle; Tuesday–Friday: BEZ235 (per os, 45 mg kg$^{-1}$ per day); Saturday and Sunday: off for 3 weeks. Tumour growth was measured every 3 days. All animal work was carried out in line with the UK Animal Scientific Procedures Act 1986 and the EU Directive 2010.

**Data availability.** The authors declare that the data supporting the findings of this study are available within the paper and its Supplementary Information files, public databases as indicated, or from the author upon request. TGCA colorectal cancer dataset used in this study is publicly available in cbioportal (www.cbioportal.org; study: TCGA, Nature 2012; case/patient set: 212 tumours with sequencing and CNA data).

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

## Acknowledgements

We thank the Bloomington Stock Center, TRiP at Harvard Medical School (NIH/NIGMS R01-GM084947), Vienna *Drosophila* RNAi Center, Volker Hartenstein, Yashi Ahmed and Andreas Wodarz for reagents; Jessica Esernio for technical assistance; Erika Bach, Gregory David, Kevan Shokat and members of the Cagan laboratory for helpful discussions. This work was supported by NIH grants R01-CA109730, R01-CA084309 and U54OD020353. O.J.S. and C.M. were funded by a CRUK core grant and an ERC investigator grant.

## Author contributions

E.B. and R.L.C. designed the project and wrote the manuscript. E.B. and A.G.S.T. performed the experiments. C.M. and O.J.S. designed and performed the mouse experiments.

## Additional information

**Competing financial interests:** The authors declare no competing financial interests.

**Publisher's note**: 

