## [Peer Review File · Nature Communications]

REVIEWERS' COMMENTS:

Reviewer #1 (Remarks to the Author):

In the revised version, Bangi and colleagues have addressed all but one of my comments. The remaining issue has to do with their conclusion that 'p-4EBP levels serve as a biomarker for this (BEZ235) sensitivity'. I suggested screening cell lines for p4EBP levels to ask if it correlates with sensitivity to BEZ235 as they conclude. In response, the authors show elevated p4EBP levels in *apc Kras* mouse tumors compared to *apc Kras Pten* tumors. Without accompanying BEZ235 sensitivity data for these tumor samples, the response goes only part way towards addressing my point. Alternatively, they could compare p4EBP levels in DLD-1 vs. DLD-1-WT cells and HCT116 vs. HCT116-WT, for which they have BEZ235 sensitivity data. The authors show data for p4EBP levels after treatment with bortezomib, but it is the baseline data I am asking for, which is more relevant as a predictive biomarker. Validating p4EBP, a surrogate for mTORC1 activity, as a predictive biomarker for BEZ235 sensitivity is important because it is a major conclusion from this work.

A minor point:

The authors describe Ras, p53, APC, and PIK3CA mutant DLD-1 cells as having a 'mutation status similar to our *Drosophila rasG12V p53Ri ptenRi apcRi* model. This is misleading because *pten* and PI3K oppose each other. What is similar is the molecular state, because both models have low downstream mTORC1 activity. This should be clarified.

Reviewer #3 (Remarks to the Author):

I restate my enthusiasm for this exciting manuscript that convincingly establishes the fly as a model to identify and characterise new mechanisms of drug resistance in multigenic tumour models. The authors have addressed most of my concerns. There are only two outstanding issues re: the following two points:

We measured the amount of food ingested by control, *rasG12V*, and *rasG12V p53Ri ptenRi apcRi* quadruple animals 4 days after the induction of transgenes and found no difference between these genotypes. This is now shown in supplementary figure 3d.

This looks fine but the methods need to state how food intake was measured (presumably CAFÉ assay) and over what time period.

The disseminated cells are loosely attached to tracheal branches and are rarely well-preserved using traditional fixation methods. We are working to develop a fixation method that will allow a detailed immunohistochemical analysis of tumor cell-trachea interactions and allow us to explore any role the trachea play in dissemination. This is an interesting topic but will require a large number of experiments to understand the mechanisms of pathfinding, etc.

This sounds reasonable but, if the authors are currently unable to provide visual evidence for the following statements, they should be removed from the manuscript (they are not central its message):

"These migrating cells commonly enwrapped tracheal branches, a tubular network that provides oxygen to *Drosophila* tissues"

"The disseminated foci were attached to the abdominal body wall or other organs through the

tracheal system"

"Migrating cells in triple and quadruple combinations were significantly larger and extended longer processes (Figure 3I) that made more extensive contacts with the overlying tracheal branches"

The editor also asked me to comment on the authors' response to Reviewer #2. You will find my comments below.

We do now include some controls that the reviewer requested (see below). Please note the reference to "critical controls" is in several instances asking for data that is unrelated to the central point of the manuscript or are not proper controls; as we discuss below, we provide the relevant controls. Adding all of the experiments discussed by Reviewer #2 would make the manuscript more dense, more confusing, and would not add significantly to our conclusions.

I overall agree with the authors' comments (see more specific comments for each of the points below).

1. We chose this assay because (i) dissemination is readily quantifiable, (ii) dissemination represents the culmination of most of the other aspects of transformation and (iii) dissemination and subsequent metastasis is the primary cause of mortality in colorectal cancer patients. Of note, our mammalian assays look at cell viability, 3D culture viability, xenograft and allograft tumor growth. That is, we use an overall wide range of assays and experimental systems to validate our conclusions.

I agree - dissemination is an integrative readout that lends itself to medium-throughput approaches, but the authors have also quantified more specific aspects of the transformation process in both flies and mammalian systems.

2. We agree this is an interesting question are in the process of exploring this question by systematically manipulating PI3K pathway components in rasG12V alone, ptenri alone and rasG12V p53ri ptenri apcri backgrounds. This represents a major project on its own; the data would not specifically address the main conclusions drawn in this manuscript. Given the already large size of the manuscript and the significant effort that will be required to explore this question, initiating an exploration of the mechanisms by which AKT protein is stabilized-reported by others as well as within this manuscript-is well beyond the scope of the work described here.

I agree.

3. These figures serve to establish the fact that Drosophila models are relevant and useful cancer drug discovery tools. Once validated, we then use these models to explore mechanisms of drug response and resistance and provide novel insights. We agree that the manuscript contains a large amount of data, but the ability to present a complete story with clinical relevance is a strength.

I agree. The main strength of the manuscript is that it both establishes the fly as a new model for studying drug responses of complex tumours and provides novel insights into mechanisms of drug resistance.

4. For the sequential treatment experiments, the key control is the two step therapy in which the order of drugs is reversed. This most directly examines the mechanism of induced addiction. We now include two new panels that show sequential treatment experiments (SC79/BEZ235 and Bortezomib/BEZ235) along with monotherapies (Fig 5c, 5f)

I believe that these considerations and new experiments address the reviewer's comment.

5. Total 4EBP antibody does not work well in flies so we were not able to do these controls. Total AKT levels were examined in the initial experiments where pAKT data are used to draw important

controls (e.g., Figure S3). For the remainder of the paper, p4EBP data is used to draw all conclusions.

This is not ideal, and I understand the reviewer's concern. However, I am not sure what else can be done about it, and the statements referring to these experiments in the manuscript (such as "p-AKT and p-4EBP levels had returned to baseline (Figure 4d,e) and (ii) and total levels of AKT protein were very low (Figure S3i)" seem justified by the current data.

6. DMSO and BEZ235 treated parental animal controls for the tsc knock down experiment are now included in Fig 4h. Parental controls for the raptor knockdown experiment (DMSO and Bortezomib/BEZ235 treated quadruple animals), are shown repeatedly in the same figure (see panels 5e and 5f). In order to minimize redundancy, we have not included the same data a third time.

This point seems to have been addressed by the data provided.

7. To simplify presentation and to present the full data, these controls are presented in Supplementary Figure 4g, h as well as discussed in the text.

Addressed.

8. As our tsc knock-down experiments in *Drosophila* demonstrated, this is not an ideal approach to test our model; knockdown experiments do not provide the temporal precision required to interpret the dynamic changes in the cells. We already show that the two step therapy is more effective than BEZ235 alone in multiple mammalian models; we feel that tsc knock-downs would not provide the relevant controls in these experiments.

This makes sense to me.

9. In vivo experiments always show some variability between biological replicates. In other words, not every animal responds or responds to the same degree in our drug feeding experiments. This is also evident in our dissemination experiments where there is animal to animal variation in the response to the two step therapy. For this reason, we scored a large number of animals in each experiment (60 for the dissemination assay and 10 guts/biological replicate in westerns). As a side note, while both doses of bortezomib show significant suppression of dissemination in combination with BEZ235 in 5e (formerly 5d) and the new panel 5f, the difference between the efficacy of the two doses is not statistically significant

This seems appropriately addressed.

10. It is stated that PTEN loss is equivalent to gaining an activated PI3K mutation, whereas we know that these effects in mammalian cells are vastly different. They should not be used interchangeably. Despite the well established differences in pten loss and PIK3CA activation, these mutations behave similarly in their response to the two step therapy. We have clarified this point in the text.

I was unable to find this (please state at least page numbers in future responses).

Reviewer #1 (Remarks to the Author):

In the revised version, Bangi and colleagues have addressed all but one of my comments. The remaining issue has to do with their conclusion that 'p-4EBP levels serve as a biomarker for this (BEZ235) sensitivity'. I suggested screening cell lines for p4EBP levels to ask if it correlates with sensitivity to BEZ235 as they conclude. In response, the authors show elevated p4EBP levels in *apc* Kras mouse tumors compared to *apc* Kras Pten tumors. Without accompanying BEZ235 sensitivity data for these tumor samples, the response goes only part way towards addressing my point. Alternatively, they could compare p4EBP levels in DLD-1 vs. DLD-1-WT cells and HCT116 vs. HCT116-WT, for which they have BEZ235 sensitivity data. The authors show data for p4EBP levels after treatment with bortezomib, but it is the baseline data I am asking for, which is more relevant as a predictive biomarker. Validating p4EBP, a surrogate for mTORC1 activity, as a predictive biomarker for BEZ235 sensitivity is

important because it is a major conclusion from this work.

We have now removed discussion of biomarkers from the manuscript.

A minor point:

The authors describe Ras, p53, APC, and PIK3CA mutant DLD-1 cells as having a 'mutation status similar to our *Drosophila* ras^{G12V} p53^{Ri} pten^{Ri} apc^{Ri} model. This is misleading because pten and PI3K oppose each other. What is similar is the molecular state, because both models have low downstream mTORC1 activity. This should be clarified.

We now restate this in the text: "The human colorectal cancer cell line DLD-1 contains mutations in Ras, p53, APC, and an activating mutation in the PI3K pathway component PIK3CA, a combination that leads to an overall molecular state similar to our *Drosophila* ras^{G12V} p53^{Ri} pten^{Ri} apc^{Ri} model."

Reviewer #3:

I restate my enthusiasm for this exciting manuscript that convincingly establishes the fly as a model to identify and characterise new mechanisms of drug resistance in multigenic tumour models. The authors have addressed most of my concerns. There are only two outstanding issues re: the following two points:

We measured the amount of food ingested by control, rasG12V, and rasG12V p53ri ptenri apcri quadruple animals 4 days after the induction of transgenes and found no difference between these genotypes. This is now shown in supplementary figure 3d.

This looks fine but the methods need to state how food intake was measured (presumably CAFÉ assay) and over what time period.

We now note in Supplemental Legend S3: “Measurements were done for a period of 8 hours, 4 days after induction of transgenes using the capillary feeding assay (CAFE).”

This is also stated in the methods now and two papers describing this method are cited.

The disseminated cells are loosely attached to tracheal branches and are rarely well-preserved using traditional fixation methods. We are working to develop a fixation method that will allow a detailed immunohistochemical analysis of tumor cell-trachea interactions and allow us to explore any role the trachea play in dissemination. This is an interesting topic but will require a large number of experiments to understand the mechanisms of pathfinding, etc.

This sounds reasonable but, if the authors are currently unable to provide visual evidence for the following statements, they should be removed from the manuscript (they are not central its message):

"These migrating cells commonly enwrapped tracheal branches, a tubular network that provides oxygen to *Drosophila* tissues"

By removing the red channel (muscle) and converting the blue channel (Laminin) into grayscale, we were able to clearly demonstrate the association between migrating cells and tracheal branches. This is now clearly shown in Figure 2p, inset. We now state in the results section:

Though rarely preserved during fixation, these migrating cells were commonly observed to enwrap tracheal branches, a tubular network that provides oxygen to *Drosophila* tissues (Figure 2p, inset)

"The disseminated foci were attached to the abdominal body wall or other organs through the tracheal system"

We now include a close up view of a region of Figure 3m that shows the association of GFP foci with tracheal branches as an inset. The text now reads:

The disseminated foci were attached to the abdominal body wall or other organs through the tracheal system (Figure 3m, inset), which presumably provided tracks for migrating cells to reach distant sites as well as a source of oxygen.

"Migrating cells in triple and quadruple combinations were significantly larger and extended longer processes (Figure 3l) that made more extensive contacts with the overlying tracheal branches"

We have removed the second part of the sentence, which now reads: "Migrating cells in triple and quadruple combinations were significantly larger and extended longer processes (Figure 4l)"

The editor also asked me to comment on the authors' response to Reviewer #2. You will find my comments below.

We especially appreciate the additional effort by Reviewer #3 to look through Reviewer #2's comments.

We do now include some controls that the reviewer requested (see below). Please note the reference to "critical controls" is in several instances asking for data that is unrelated to the central point of the manuscript or are not proper controls; as we discuss below, we provide the relevant controls. Adding all of the experiments discussed by Reviewer #2 would make the manuscript more dense, more confusing, and would not add significantly to our conclusions.

I overall agree with the authors' comments (see more specific comments for each of the points below).

1. We chose this assay because (i) dissemination is readily quantifiable, (ii) dissemination represents the culmination of most of the other aspects of transformation and (iii) dissemination and subsequent metastasis is the primary cause of mortality in colorectal cancer patients. Of note, our mammalian assays look at cell viability, 3D culture viability, xenograft and allograft tumor growth. That is, we use an overall wide range of assays and experimental systems to validate our conclusions.

I agree - dissemination is an integrative readout that lends itself to medium-throughput approaches, but the authors have also quantified more specific aspects of the transformation process in both flies and mammalian systems.

2. We agree this is an interesting question and are in the process of exploring this question by systematically manipulating PI3K pathway components in rasG12V alone, pten^{ri} alone and rasG12V p53^{ri} pten^{ri} apcri backgrounds. This represents a major project on its own; the data would not specifically address the main conclusions drawn in this manuscript. Given the already large size of the manuscript and the significant effort that will be required to explore this question, initiating an exploration of the mechanisms by which AKT protein is stabilized- reported by others as well as within this manuscript- is well beyond the scope of the work described here.

I agree.

3. These figures serve to establish the fact that Drosophila models are relevant and useful cancer drug discovery tools. Once validated, we then use these models to explore mechanisms of drug response and resistance and provide novel insights. We agree that the manuscript contains a large amount of data, but the ability to present a complete story with clinical relevance is a strength.

I agree. The main strength of the manuscript is that it both establishes the fly as a new model for studying drug responses of complex tumours and provides novel insights into mechanisms of drug resistance.

4. For the sequential treatment experiments, the key control is the two step therapy in which the order of drugs is reversed. This most directly examines the mechanism of induced addiction. We now include two new panels that show sequential treatment experiments (SC79/BEZ235 and Bortezomib/BEZ235) along with monotherapies (Fig 5c, 5f)

I believe that these considerations and new experiments address the reviewer's comment.

5. Total 4EBP antibody does not work well in flies so we were not able to do these controls. Total AKT levels were examined in the initial experiments where pAKT data are used to draw important controls (e.g., Figure S3). For the remainder of the paper, p4EBP data is used to draw all conclusions.

This is not ideal, and I understand the reviewer's concern. However, I am not sure what else can be done about it, and the statements referring to these experiments in the manuscript (such as "p-AKT and p-4EBP levels had returned to baseline (Figure 4d,e) and (ii) and total levels of AKT protein were very low (Figure S3i)" seem justified by the current data.

6. DMSO and BEZ235 treated parental animal controls for the tsc knock down experiment are now included in Fig 4h. Parental controls for the raptor knockdown experiment (DMSO and Bortezomib/BEZ235 treated quadruple animals), are shown repeatedly in the same figure (see panels 5e and 5f). In order to minimize redundancy, we have not included the same data a third time.

This point seems to have been addressed by the data provided.

7. To simplify presentation and to present the full data, these controls are presented in Supplementary Figure 4g, h as well as discussed in the text.

Addressed.

8. As our tsc knock-down experiments in *Drosophila* demonstrated, this is not an ideal approach to test our model; knockdown experiments do not provide the temporal precision required to interpret the dynamic changes in the cells. We already show that the two step therapy is more effective than BEZ235 alone in multiple mammalian models; we feel that tsc knock-downs would not provide the relevant controls in these experiments.

This makes sense to me.

9. In vivo experiments always show some variability between biological replicates. In other words, not every animal responds or responds to the same degree in our drug feeding experiments. This is also evident in our dissemination experiments where there is animal to animal variation in the response to the two step therapy. For this reason, we scored a large number of animals in each experiment (60 for the dissemination assay and 10 guts/biological replicate in westerns). As a side note, while both doses of bortezomib show significant suppression of dissemination in combination with BEZ235 in 5e (formerly 5d) and the new panel 5f, the difference between the efficacy of the two doses is not statistically significant

This seems appropriately addressed.

10. It is stated that PTEN loss is equivalent to gaining an activated PI3K mutation, whereas we know that these effects in mammalian cells are vastly different. They should not be used interchangeably. Despite the well established differences in pten loss and PIK3CA activation, these mutations behave similarly in their response to the two step therapy. We have clarified this point in the text.

I was unable to find this (please state at least page numbers in future responses).

In the final paragraph of the Discussion we now state: “Furthermore, the two-step therapy was also effective in cell lines that carry activating mutations in PIK3CA, rather than PTEN loss. Despite the well established differences in pten loss and PIK3CA activation, these mutations behave similarly in their response to the two step therapy, suggesting that tumors carrying either PTEN loss or oncogenic mutations in PIK3CA could benefit from the two-step therapy.”